# WISE: A Long-Horizon Agent in Minecraft with Why-Which Reasoning

## Abstract

Rapid advances have been made in developing general-purpose embodied agent in environments like Minecraft through the adoption of LLM-augmented hierarchical approaches. Despite their promise, low-level controllers often become performance bottlenecks due to repeated execution failures. We argue that a key limitation is not only the lack of episodic memory, but also the decoupling of *what-where-when* memory from *which-why* reasoning. To address this, we propose **WISE** (Which-Why Informed Semantic Explorer), a long-horizon agent framework with an enhanced low-level controller equipped with a Causal Event Graph that augments episodic memory with explicit causal structure linking observations to task relevance. Unlike prior work such as MrSteve, which relies on feature similarity for retrieval, WISE enables robust recall under viewpoint changes and supports opportunistic task reordering through causal reasoning. Building on this memory, we propose an Opportunistic Task Scheduler that dynamically re-prioritizes subtasks when causally relevant opportunities are detected. We further equip WISE with a multi-scale progressive exploration strategy to provide spatially comprehensive observations for downstream reasoning. Experiments show that WISE largely improves task success and efficiency on long-horizon sparse tasks, particularly in settings requiring adaptive decision-making.

## 1 Introduction

Building generally capable embodied agents that can solve long-horizon tasks in complex, open-ended environments remains a fundamental challenge in artificial intelligence. Minecraft has emerged as a prominent benchmark for studying this problem because it presents a procedurally generated world with rich interaction dynamics requiring exploration, resource acquisition, tool crafting, and long-term planning Guss et al. (2019); Fan et al. (2022); Baker et al. (2022). Even seemingly simple objectives, such as *"obtain beef"*, require multi-step reasoning: the agent must first infer the semantic link between beef and cows, then explore sparse biomes to locate cattle, navigate complex terrain, engage in interaction or combat, and finally collect the resulting resources. Failures at any stage (e.g., losing track of the target entity) often lead to costly re-exploration. More complex tasks, such as diamond acquisition, can require tens of thousands of environment steps Lifshitz et al. (2023), making reinforcement learning from scratch prohibitively difficult due to sparse rewards and vast exploration spaces.

Recent progress has shown that Large Language Model (LLM)-augmented hierarchical frameworks provide a promising solution Wang et al. (2023); Zhou et al. (2024). These systems decompose long-horizon tasks into two levels: a high-level planner that leverages LLM reasoning to generate subgoals and a low-level controller that executes them. This decomposition largely reduces task complexity and has enabled notable advances in embodied decision making. For such frameworks to succeed, however, high-level planning and low-level execution must improve jointly. Existing research has primarily concentrated on enhancing high-level planning through skill libraries Zhu et al. (2023); Wang et al. (2023); Qin et al. (2024), multimodal experience repositories Li et al. (2024), or increasingly powerful language reasoning modules.

In contrast, comparatively little attention has been paid to the low-level controller, which often becomes the dominant bottleneck in practice Cai et al. (2023). Existing methods frequently assume that once a subgoal

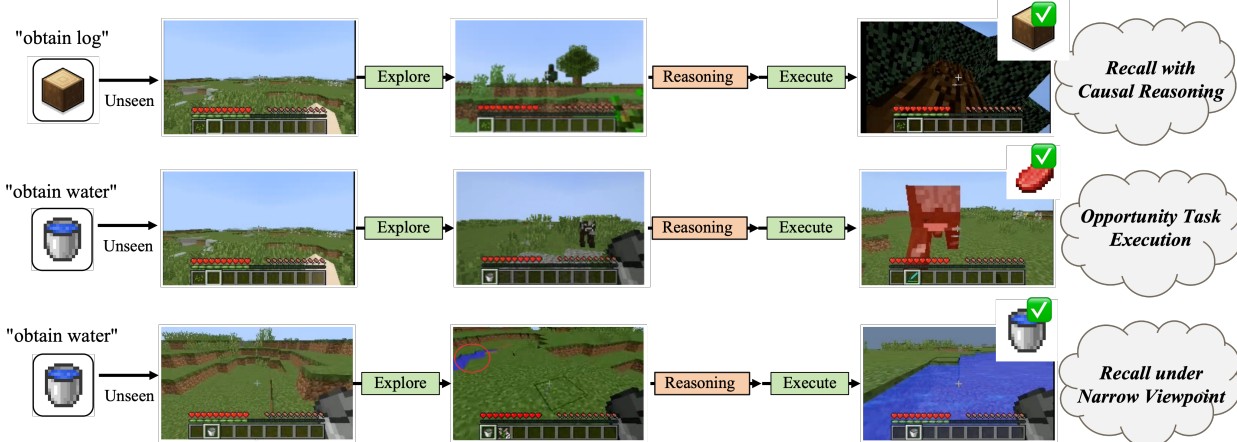

Figure 1: Comparison of WISE and prior approaches across three key capabilities. **Causal Memory Recall:** WISE recalls previously observed cows for the task "obtain beef" by leveraging explicit causal knowledge (cow → beef), whereas MrSteve relies solely on visual similarity. **Opportunistic Task Execution:** WISE dynamically reorders the task queue to exploit chance encounters (e.g., immediately killing a nearby cow), while prior approaches rigidly follow a predefined task sequence. **Robust Recall under Viewpoint Changes:** WISE employs semantic entity representations that are invariant to viewpoint variations, enabling consistent recognition of cows from different perspectives; prior approaches depend on raw visual features and often fail under viewpoint changes. Green check marks indicate successful behavior.

is generated, the controller can reliably and efficiently execute it—an assumption that often breaks down in large, sparse environments. A recent attempt to address this limitation is MrSteve Park et al. (2025), which augments Steve-1 with Place Event Memory (PEM). PEM organizes past observations according to *what* happened, *where* it occurred, and *when* it was observed, enabling retrieval of previously encountered locations and events. For example, an agent may remember that it previously observed a cow in a forest region and later navigate back to that location.

However, we argue that the primary limitation is not only memory capacity itself, but also the absence of semantic and causal reasoning over memory. Existing systems Wang et al.; Zhu et al. (2023); Park et al. (2025) isolate episodic recall from decision making. PEM retrieves observations using cosine similarity between MineCLIP visual features and task embeddings, resulting in two key deficiencies. First, retrieval based on raw visual similarity is semantically brittle: changes in viewpoint, occlusion Cai et al. (2023); Zhou et al. (2024), or environmental conditions frequently cause memory recall to fail. Second, PEM stores observations without representing their causal implications. The memory may record that a cow was previously observed, but it cannot infer why this observation matters—namely, that cows can provide beef. Consequently, when the future task becomes *"obtain beef"*, the agent lacks a mechanism to identify previously encountered cows as relevant opportunities.

More fundamentally, existing controllers Lifshitz et al. (2023); Park et al. (2025) execute subgoals according to fixed action sequences and cannot answer the question: *which action should be performed right now given newly available information?* Consider an agent traveling to collect wood that unexpectedly encounters a cow. An ideal agent should immediately recognize that the encounter creates an opportunity to satisfy a future subgoal. Existing systems Qin et al. (2024); Yuan et al.; Li et al. (2025) instead continue executing the prescribed sequence and return later, incurring unnecessary navigation cost. Exploration, memory, and planning operate as disconnected modules; consequently, the overall system fails to exploit the synergy among them.

In this work, we introduce **WISE** (**W**hich-**W**hy **I**nformed **S**emantic **E**xplorer), a long-horizon agent framework that closes the loop between exploration, memory, and decision making. As illustrated in Figure 1, WISE extends conventional episodic memory with explicit semantic and causal structure, enabling the agent not only to remember *what*, *where*, and *when*, but also to understand *why* an observation matters and *which*

future tasks it may enable. The core of WISE is a *Causal Event Graph*, a semantic memory structure that augments episodic observations with explicit causal relationships extracted by a Vision-Language Model. Instead of storing observations as isolated visual memories, the graph links entities and downstream task outcomes through causal edges (e.g., *cow* $\rightarrow$ `CAN_OBTAIN` $\rightarrow$ *beef*). This semantic layer enables causally grounded retrieval and transforms memory from passive storage into actionable knowledge. Building upon this representation, we introduce an *Opportunistic Task Scheduler* that continuously reasons over causal memories and dynamically reprioritizes pending subtasks. Rather than rigidly following a predefined execution sequence, the scheduler adapts to newly observed opportunities. When a causally relevant entity appears, WISE immediately updates its task priorities and exploits the opportunity online, thereby avoiding redundant navigation and improving long-horizon efficiency. Finally, to support this reasoning process with sufficiently rich observations, we introduce a multi-scale progressive exploration strategy that efficiently expands environmental coverage while minimizing revisitation. Together, exploration, memory, and scheduling form a closed-loop architecture: exploration acquires observations, memory converts observations into causal knowledge, and planning consumes this knowledge for adaptive decision making. Extensive experiments in large-scale Minecraft environments demonstrate the effectiveness of WISE. Compared with the current state-of-the-art low-level controller MrStevePark et al. (2025), WISE achieves a 14% improvement in exploration coverage, a 30% increase in sequential sparse task success with 26.4% lower completion time, and a 44% increase in adaptive non-sequential task success with 42.5% less completion time. Ablation studies further reveal that WISE's full-model performance significantly exceeds the sum of individual module improvements, suggesting strong synergy among *Causal Event Graph*, *Oppotunistic Task Scheduler*,and *Multi-scale Progressive Exploration* components.

Our contributions are as follows.

1. We identify a key but underexplored limitation in long-horizon embodied agents: the disconnect between episodic memory and causal decision-making. We argue that an effective low-level controller must go beyond remembering *what*, *where*, and *when*, and instead reason about *why* observations matter and *which* future actions they enable.

2. We propose **WISE**, a unified embodied framework that closes the loop between exploration, memory, and planning. Its core is a Causal Event Graph that augments episodic memory with VLM-derived semantic and causal relations, enabling robust retrieval and causally grounded reasoning.

3. We introduce an Opportunistic Task Scheduler that dynamically reprioritizes subtasks based on causal relevance, transforming the low-level controller from a rigid executor into an adaptive decision-maker capable of exploiting newly emerging opportunities.

4. We present a multi-scale progressive exploration strategy that improves coverage efficiency while providing the reasoning module with spatially comprehensive observations for long-horizon decision making.

## 2 Related work

### 2.1 Low-Level Control in Minecraft

Early works trained policy models for simple Minecraft tasks Guss et al. (2019). VPT Baker et al. (2022) showed that vision-to-action mappings can be learned from unlabeled online videos at scale. Steve-1 Lifshitz et al. (2023) extended VPT by conditioning on text instructions. GROOT Cai et al. (2024) used reference videos instead of text for goal-conditioned control. MineDreamer Zhou et al. (2024) leveraged Steve-1 to generate subgoal images for better execution.

More recent research focuses on increasing the architectural complexity and adaptability of controllers. The STEVE Series Zhao et al. (2024) systematically explored the interplay between planning granularity and observation encoding. Odyssey Liu et al. (2025) and LARM Li et al. (2025) introduced open-world skill libraries and auto-regressive models to bridge the gap between reactive control and long-horizon deliberative planning. ADAM Yu & Lu and Steve-Evolving Xie et al. (2026) represent the cutting edge of this direction:

the former constructs a causal graph through interaction, while the latter distills execution diagnoses into reusable guardrails and skills. However, these systems often treat memory as a high-level knowledge store rather than an integrated component of low-level execution. MrSteve Park et al. (2025) introduced Place Event Memory (PEM) to provide the low-level controller with episodic context, but its retrieval relies on raw visual similarity. WISE addresses this by extending PEM into a Causal Event Graph, enabling the controller to reason about why a memory is relevant to future tasks.

## 2.2 Memory In Agents

Memory is essential for long-horizon tasks, and many studies have explored different storage and retrieval mechanisms for embodied agents. Sumers et al. (2023) propose a framework that categorizes agent memory into three types: in-context, external, and in-weights. They highlight the trade-off between retrieval accuracy and update speed, which directly motivates the two-level retrieval design in WISE.

At the plan level, several memory systems have been developed. Voyager Wang et al. stores reusable skill programs across episodes, enabling lifelong learning in Minecraft without human intervention. GITM Zhu et al. (2023) uses structured world knowledge to improve goal decomposition for technology-tree tasks. JARVIS-1 Wang et al. (2024) and MP5 Qin et al. (2024) store both multimodal observations and successful plans for situation-aware retrieval, with JARVIS-1 showing that multimodal goal specification greatly improves retrieval precision. Optimus-1 Li et al. (2024) introduces a hybrid memory pool that summarizes visual, textual, and action trajectories, achieving strong results on long-horizon benchmarks.

Despite these advances, most existing memory systems focus on high-level planning—they store which plan worked, not why a specific low-level observation matters. MrSteve's PEM is the first memory system designed for low-level control, but it still retrieves memories based on raw visual features. WISE fills this gap by constructing a Causal Event Graph that connects low-level observations to high-level goals through explicit causal edges. A related system, Yu & Lu also builds a causal graph via interaction, but it targets high-level knowledge accumulation and does not integrate causal memory with a dynamic low-level scheduler.

## 2.3 Exploration in Open Worlds

Exploration strategies span several paradigms, each with a characteristic failure mode. Count-based methods Tang et al. (2017); Chang et al. (2024) encourage novelty but accumulate coverage gaps super-linearly with map size. Frontier-based methods Yamauchi (1998); Sun et al. (2025) improve boundary awareness but ignore large unvisited interior regions. NovelCraft Feeney et al. (2023) further showed that existing exploration policies fail to detect and adapt to novel objects in procedurally generated worlds, underscoring the need for multi-scale hierarchical planning that balances global coverage and local detail. VPT-Nav Park et al. (2025)) adapts to local dynamics but is reactive rather than globally planned. Active information-gain methods Chaplot et al. (2020a;b); Zhang et al. (2021) are theoretically well-motivated but computationally intractable under Minecraft's $\leq 20$ ms per-step real-time constraint. WISE's multi-scale strategy is the first to synthesise all three strengths in minecraft: $O(\log n)$ global efficiency from quadtree indexing, boundary-awareness from frontier scoring, and a local completeness guarantee from Voronoi decomposition.

# 3 Method

In this section, we introduce our proposed long-horizon embodied agent, WISE. We first describe the problem setting and then present the construction of each module.

**Problew setting**

We consider sparse long-horizon task scenarios in which an embodied agent continuously receives a sequence of tasks $\{\tau_n\}_{n=1}^{\infty}$ via textual instructions (e.g., *"Obtain beef"*) generated either directly by the environment or decomposed by a large language model (LLM). We assume that task-relevant resources occur infrequently and are sparsely distributed throughout large environments. Consequently, successful task completion requires the agent not only to memorize observations from explored regions but also to semantically understand their relevance to future tasks.

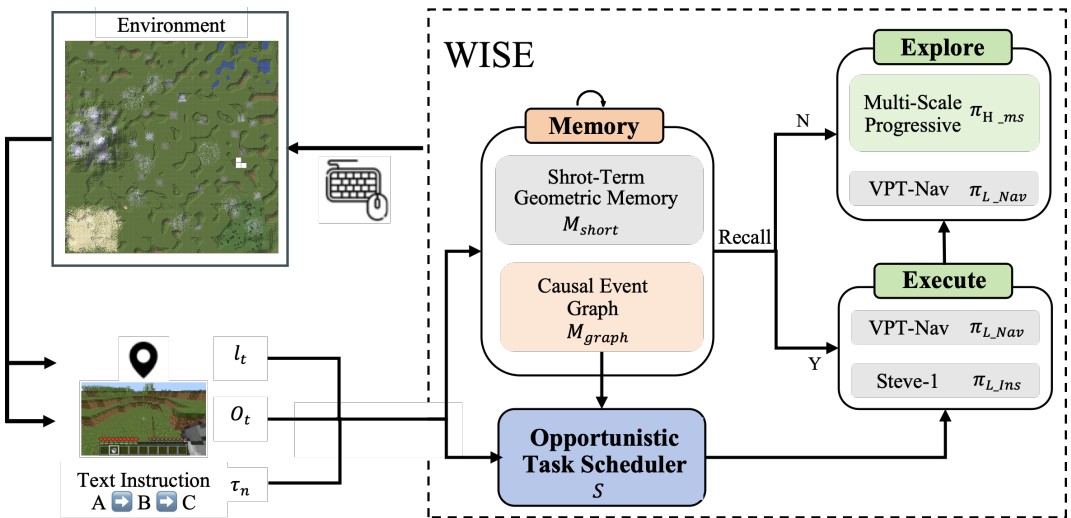

Figure 2: Overall architecture of WISE. Given a text instruction (e.g., "A → B → C"), the agent alternates among three phases: *Explore*, *Reason*, and *Execute*. During exploration, a multi-scale progressive policy systematically explores the environment and stores observations in two memory systems: a short-term geometric memory for efficient retrieval of recent observations and a Causal Event Graph for storing semantic entities and causal relationships. During execution, the agent uses VPT-Nav for goal-directed navigation and Steve-1 for low-level task execution. The Opportunistic Task Scheduler dynamically reorders subtasks according to causally relevant memories retrieved from the graph.

At the beginning of each episode, at every timestep $t$, the agent receives an observation:

$$X_t = \{O_t, l_t, t\},$$

where $O_t = i_t \in \mathbb{R}^{H \times W \times C}$ denotes the first-person RGB observation and

$$l_t = (coord_x, coord_y, coord_z, yaw, pitch) \in \mathbb{R}^5$$

represents positional information, including the agent's 3D coordinates and camera orientation relative to the initial frame of reference.

**Framework Overview**

WISE is a memory-augmented agent framework consisting of three major components: a *Memory Module*, a *Solver Module*, and an *Opportunistic Task Scheduler*. The memory module $M_t$ contains two complementary memory structures. First, a *Causal Event Graph*, denoted as $M_{\mathrm{graph}}$, extends conventional episodic memory with semantic and causal knowledge. Second, a *short-term geometric memory*, $M_{\mathrm{short}}$, provides low-latency retrieval of recently observed experiences.

Based on retrieved memory contents, a mode selector within the solver module dynamically switches between *Explore* and *Execute* modes. Together with a multi-scale progressive exploration policy, these modules form the complete WISE framework. Algorithm 1 outlines the task-solving loop of WISE. By jointly exploring, memorizing, and reasoning over causal relationships, WISE enables efficient decision-making for sparse long-horizon tasks. In the following sections, we describe each module in detail, beginning with the memory module.

## 3.1 Causal Event Graph for Semantic Episodic Memory

Episodic memory in embodied agents typically records *what* happened, *where* it happened, and *when* it occurred. However, for long-horizon tasks, this information alone is insufficient. An intelligent agent must

---

**Algorithm 1** WISE Single Loop

---

**Require:** Memory $M_t = (M_{short}, M_{graph})$, Task $\tau = (\tau_1, \tau_2, ... \tau_t ... \tau_n)$ and Opportunistic Task Scheduler S.
 1: $candidates = (candidate_1, candidate_2) \leftarrow \text{Read}(M_t, \tau_t)$
 2: $S(\tau_t + i), l_{t+i} \leftarrow \text{Read}(M_{graph}, \tau, S)$
 3: **if** $candidates \neq \emptyset$ **then**
 4:      $candidate = score(candidates)$
 5:      $X_t, l_t = \text{OneOf}(candidate)$
 6:      Navigate to $l_t$ with $\pi_{\text{L-Nav}}$
 7:      Execute $\tau_n$ with $\pi_{\text{Inst}}$
 8: **else if** $S(\tau_t + i) \neq \emptyset$ **then**
 9:      Reprioritizing tasks $\tau = (\tau_1, \tau_2, ... \tau_t + i, \tau_t ... \tau_n)$
10:      Navigate to $l_{t+i}$ with $\pi_{\text{L-Nav}}$
11:      Execute $\tau_n$ with $\pi_{\text{Inst}}$
12: **else**
13:      Explore with $\pi_{H_M s}$, $\pi_{L_N av}$
14: **end if**

---

additionally understand *why* a past event matters for its current objectives and *which* future tasks may be enabled by newly encountered entities. Existing memory systems, including Place Event Memory (PEM), primarily focus on the former: storing observations and their associated spatio-temporal context. The proposed Causal Event Graph extends this paradigm by explicitly encoding causal relationships among entities and tasks. This transforms episodic experiences into structured semantic knowledge that supports opportunistic reasoning and dynamic replanning.

### 3.1.1 Short-Term Geometric Memory

As shown in Figure 3, given Minecraft game frames, a MineCLIP encoder first extracts visual representations. DP-Means Dinari & Freifeld (2022) clustering is then applied to group temporally related observations into event clusters. Subsequently, cosine similarity between the current task embedding and each event-cluster center is computed. Clusters whose similarity exceeds a predefined threshold are ranked, and the top-$K$ clusters are retrieved as memory cues. This mechanism follows the memory design adopted in MrSteve but lacks explicit causal reasoning capabilities.

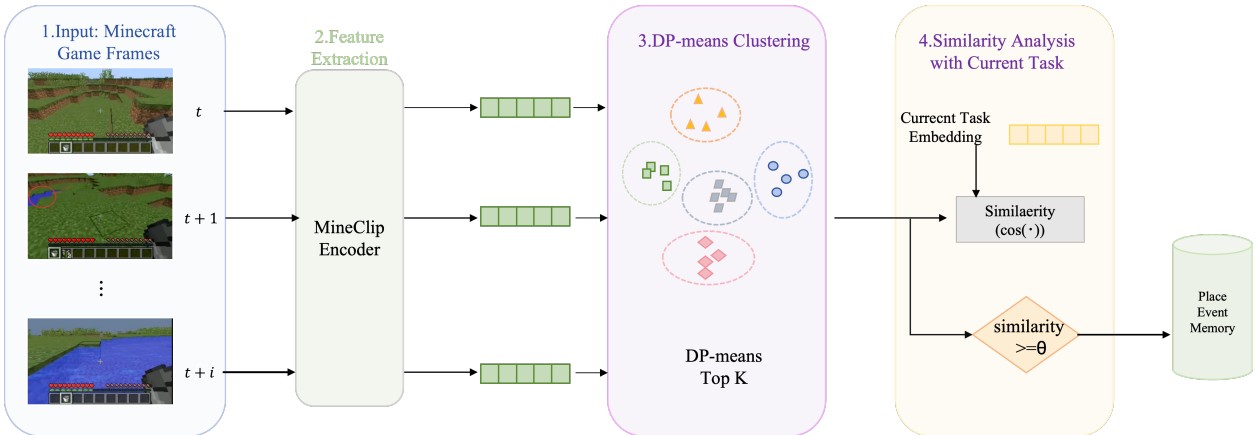

Figure 3: Short-Term Geometric Memory. Given Minecraft video frames, a MineCLIP encoder extracts visual embeddings, which are grouped into event clusters using DP-means clustering. The system then computes cosine similarity between the current task embedding and each cluster centroid. Clusters with similarity above a predefined threshold are ranked, and the top-$K$ matches are retrieved as memory cues.

To enable rapid and low-latency retrieval of recent experiences, WISE maintains a short-term memory structure identical to PEM. Observation embeddings are computed as

$$e_t = Enc_v(i_{t-H:t}),$$

where MineCLIP extracts representations over a temporal window of size $H = 16$. Experience frames are represented as

$$x_t = \{e_t, l_t, t\},$$

and organized into place clusters based on the agent's 2D position and yaw orientation. Within each place cluster, event clusters are further formed via DP-Means clustering on video embeddings.

When memory capacity is exceeded, the oldest frame from the largest event cluster is removed. This memory layer efficiently answers *where* and *when* queries. However, its representation of *what* remains restricted to visual similarity alone: it captures appearance but lacks semantic understanding.

### 3.1.2  *VLM-Driven Causal Graph Construction*

To understand *why* a particular observation matters, we construct a semantic knowledge graph on top of the geometric memory layer. As shown in Figure 4, the graph construction process begins with keyframe selection. We hypothesize that informative keyframes should preserve both semantic content and visual diversity from the original observation stream. To this end, we propose a hybrid keyframe extraction strategy that combines clustering-based and entropy-based approaches.

Specifically, a subset of keyframes is obtained from feature clusters in the short-term geometric memory, preserving representative observations across different scenes and events. In parallel, an image entropy-based criterion selects visually informative frames with potentially rich semantic content. The two sets are merged and filtered to remove redundancy, producing a compact yet semantically diverse set of candidate keyframes. The selected keyframes are then asynchronously processed by a Vision-Language Model (VLM). Since VLM inference is computationally expensive, it is executed independently of the real-time control loop and therefore does not interrupt online decision-making.

For each keyframe, the VLM performs three operations. First, it conducts **entity extraction** to identify semantically meaningful elements in the scene, including animals, resources, terrain structures, and environmental objects. These entities are added as graph nodes and associated with the spatial coordinates of their originating observations. Second, the model constructs **causal relations**. For each extracted entity, the VLM infers downstream task-relevant outcomes that may be obtained or enabled by interacting with it. For example, a *cow* entity is connected to *beef* and *leather* via `CAN_OBTAIN` relations, while a *tree* is connected to *logs* and *leaves*. These edges explicitly encode the rationale underlying the *why* dimension: observing a cow is meaningful not because of its visual appearance, but because it implies that beef can later be acquired. Finally, the VLM captures **spatial co-occurrence** relationships by connecting entities that frequently appear together through `CO_OCCURS_WITH` edges. Such relations encode environmental regularities and exploit the observation that resources often occur in correlated spatial configurations. Together, these operations transform low-level episodic observations into structured semantic knowledge suitable for reasoning and planning.

### 3.1.3  *Two-Level Retrieval with Causal Augmentation*

When the agent receives a new task $\tau_n$, memory retrieval proceeds at two complementary levels. First, the short-term geometric layer computes cosine similarity between the task embedding $Enc_t(\tau_n)$ and stored observation embeddings, returning the top-$k$ visually matched memories. Second, the semantic layer queries the Causal Event Graph. The task description is matched against entity nodes, after which causal edges are traversed to identify observations that are semantically relevant. For example, a query such as *"obtain beef"* first matches the *beef* node and then follows incoming `CAN_OBTAIN` edges to associated *cow* instances, thereby retrieving their stored spatial locations. This causal retrieval process explains *why* a memory is selected. A cow observation is retrieved for a beef-related task not because its visual appearance resembles beef, but because the graph explicitly encodes their causal relationship.

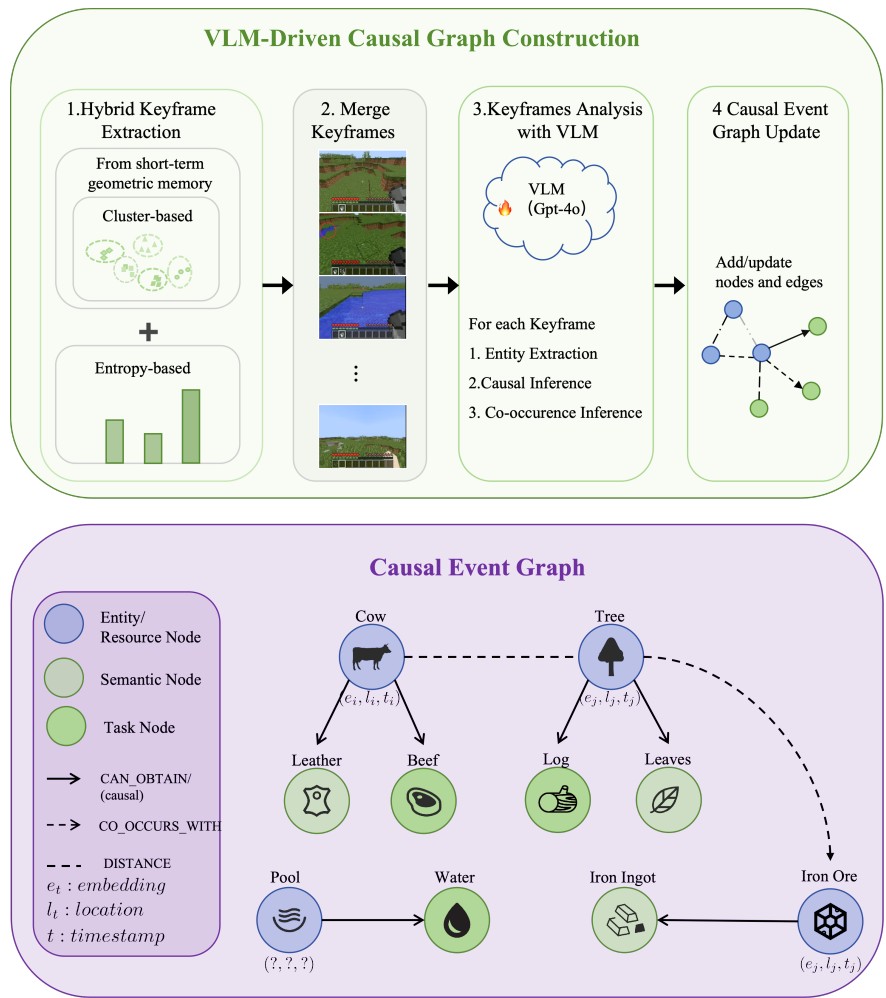

Figure 4: VLM-driven construction of the Causal Event Graph. Hybrid keyframe extraction first selects representative observations from short-term geometric memory using complementary clustering-based and entropy-based strategies, followed by redundancy reduction through keyframe merging. A Vision-Language Model (GPT-4o) then analyzes each keyframe to extract semantic entities, infer causal relationships (e.g., "cow → beef"), and identify spatial co-occurrence patterns. The resulting information is used to update the Causal Event Graph by inserting or modifying entity nodes and causal edges. This process transforms raw episodic observations into structured semantic knowledge that supports downstream reasoning and planning.

The outputs from both retrieval pathways are integrated using a weighted scoring function:

$$Score(x, \tau_n) = \lambda \cdot sim(e_x, Enct(\tau_n)) + (1 - \lambda) \cdot CausalMatch(x, \tau_n) \tag{1}$$

where $CausalMatch(x, \tau_n)$ equals 1 if observation $x$ contains an entity causally linked to task $\tau_n$ within the graph, and 0 otherwise.

This formulation provides robustness when visual similarity becomes unreliable due to viewpoint changes, occlusions, or lighting variations. Even when appearance-based retrieval fails, causal relationships provide an independent semantic retrieval pathway. This two-level design substantially improves memory robustness beyond conventional systems such as PEM.

### 3.2 Opportunistic Task Scheduler for Adaptive Task Selection

Given multiple pending subtasks, continuously evolving observations, and a set of causally structured memories, the central question becomes: *which* subtask should be executed at the current moment?

Existing systems such as MrSteve rely entirely on a fixed task sequence generated by a high-level planner. The low-level controller executes subgoals strictly according to the prescribed order and cannot deviate even when the environment presents opportunities that would make future subtasks easier to accomplish. This rigid strategy often results in inefficient behavior and missed opportunities. For example, suppose an agent encounters a cow while traveling to collect wood. Under a fixed execution order, the agent must first finish the wood-collection task and later return to the same location to obtain beef. Such behavior introduces unnecessary and redundant movement.

To address this limitation, we introduce an Opportunistic Task Scheduler that dynamically prioritizes subtasks according to current observations and retrieved causal memories. At each decision step, the scheduler constructs a unified decision context from three information sources: the current visual observation $i_t$, the set of pending subgoals $\{\tau_i\}$, and retrieved memories from the Causal Event Graph. Each pending subgoal is associated with a memory cue consisting of its highest-ranked retrieved observation, corresponding spatial coordinates, and the causal relationships connecting the memory to the task.

Each subgoal $\tau_i$ receives a composite priority score:

$$\text{Priority}(\tau_i, t) = w_u \cdot \text{Urgency}(\tau_i, t) + w_r \cdot \text{CausalRel}(\tau_i, t) + w_n \cdot [1 - \text{NavCost}(\tau_i, t)] \tag{2}$$

where each component captures a distinct decision factor. $Urgency(\tau_i, t)$ reflects temporal importance and whether the task lies on a critical execution path. $NavCost(\tau_i, t) \in [0, 1]$ estimates the normalized travel cost required to reach the target location. The key component is $CausalRel(\tau_i, t)$, which serves as the primary driver of the *which* decision. It measures whether an active causal path exists between current observations or recently retrieved memories and subgoal $\tau_i$. For instance, when the agent detects a cow and the graph contains the causal relation:

$$cow \rightarrow CAN\_OBTAIN \rightarrow beef,$$

the causal relevance score for the task "obtain beef" becomes high; otherwise, it remains near zero.

We empirically set the weights as $w_u = 0.3$, $w_r = 0.5$, and $w_n = 0.2$, reflecting the design principle that causal relevance should dominate adaptive task selection. After scoring, the scheduler reorders the pending task queue according to descending priority. If the highest-priority task differs from the currently active one, execution is preempted and control switches to the new task. This strategy is opportunistic because it exploits favorable environmental encounters that fixed plans ignore. It is also causal because task switching is driven entirely by activated semantic relationships in the knowledge graph.

**Illustrative Example.** Consider an agent executing the task sequence:

$$[\texttt{find water} \rightarrow \texttt{collect logs} \rightarrow \texttt{obtain beef}]$$

While navigating toward water, the agent observes a cow. The Causal Event Graph activates the relation:

$$cow \rightarrow CAN\_OBTAIN \rightarrow beef.$$

Consequently, the scheduler recomputes task priorities, causing the causal relevance of "obtain beef" to increase significantly. The reordered execution queue becomes:

$$[\texttt{obtain beef} \rightarrow \texttt{find water} \rightarrow \texttt{collect logs}]$$

The agent immediately acquires beef and subsequently resumes the remaining objectives, avoiding a future return trip. By contrast, a fixed-sequence policy such as MrSteve would ignore the encountered cow and revisit the same location later, resulting in unnecessary navigation cost.

### 3.3 Multi-Scale Progressive Exploration

The Causal Event Graph and Opportunistic Task Scheduler constitute the reasoning core of WISE; however, their effectiveness fundamentally depends on access to sufficiently diverse and spatially complete observations. When no task-relevant memory can be retrieved and no exploitable opportunity is identified by the scheduler, the agent must actively explore the environment to acquire new information.

To address this challenge, WISE employs a *multi-scale progressive exploration* strategy designed to efficiently cover large environments while minimizing redundant exploration. The strategy consists of three hierarchical tiers operating at different spatial resolutions. Each tier addresses limitations left unresolved by the preceding level, enabling efficient and comprehensive environmental coverage.

**Global Exploration Tier.** At the coarsest level, the global exploration tier determines which large-scale region should be visited next. We partition the environment using a quadtree representation and assign each region an exploration utility score that balances coverage gain and navigation cost. Regions containing finer-grained unexplored areas receive higher rewards, while distant regions incur larger penalties. This mechanism prevents the agent from repeatedly revisiting local neighborhoods and encourages broad spatial coverage.

**Regional Exploration Tier.** After arriving at a selected region, the agent switches to regional exploration. Specifically, frontier points—defined as boundaries separating explored and unexplored areas—are extracted from the local occupancy map. Each frontier is evaluated according to three criteria: expected coverage gain, information novelty, and navigation cost. The frontier with the highest utility is selected as the next exploration target. This strategy continuously drives the agent toward regions likely to reveal new observations and resources.

**Local Completion Tier.** Although the global and regional stages efficiently expand explored territory, small unexplored gaps may remain inside previously visited regions. To eliminate these residual blind spots, WISE introduces a local completion stage based on Voronoi decomposition. The local tier identifies interior coverage gaps and guides the agent toward these neglected areas. This process progressively fills sparse regions and ensures complete exploration of all reachable space.

Together, these three levels form a coarse-to-fine exploration framework. The global tier promotes broad spatial coverage, the regional tier expands exploration boundaries, and the local tier guarantees coverage completeness. Through this progressive design, WISE efficiently acquires the observations required for memory construction and causal reasoning in sparse long-horizon environments.

## 4 Experiments

In this section, we comprehensively evaluate WISE across four complementary dimensions. Section 4.1 describes the experimental protocol and evaluation setup. Section 4.2 evaluates large-scale exploration performance to assess the effectiveness of the proposed multi-scale exploration strategy. Section 4.3 investigates sparse sequential (ABA-Sparse) and non-sequential (ABC-Sparse) task completion, jointly evaluating memory retrieval and adaptive planning capabilities. Section 4.4 presents controlled ablation studies to isolate the contribution of each component. Additional implementation details are provided in Appendix A. All experiments are conducted in Minecraft Java Edition through the MineRL platform.

### 4.1 Experiment Setups

**Hardware and software environment.** All experiments are conducted on a single server equipped with four NVIDIA RTX 4080 GPUs running Ubuntu 22.04 LTS. The software stack includes Minecraft

Java Edition v1.11.2, MineRL v0.4.4, and the official MineCLIP checkpoint from MineDojo. GPT-4o (`gpt-4o-2024-05-13`) is used as the Vision-Language Model for semantic graph construction.

**Baseline Selection.** We compare WISE against the following representative baselines:

- *Steve-1* Lifshitz et al. (2023) is the most widely adopted low-level instruction-following controller built upon the Video Pre-Training (VPT) framework Baker et al. (2022). Following prior work Cai et al. (2024); Zhou et al. (2024), we additionally provide Steve-1 with the instruction "Go Explore" to assess its exploration capability.

- *MrSteve* Park et al. (2025) is the current state-of-the-art low-level controller based on Place Event Memory (PEM). It stores *what–where–when* information and retrieves memories using visual similarity. MrSteve additionally employs a hierarchical exploration strategy consisting of a count-based high-level goal selector and a VPT-Nav low-level controller.

- *WISE (PEM+Graph)* is an ablation variant that retains the proposed Causal Event Graph while removing multi-scale exploration and the Opportunistic Task Scheduler. This variant isolates the contribution of semantic memory without adaptive planning or enhanced exploration.

- *WISE (Quadtree-Based)* is another ablation variant that uses only the global quadtree exploration tier while removing frontier-based regional exploration and Voronoi-based local completion.

We explicitly exclude API-privileged agents, including Voyager Wang et al., JARVIS-1 Wang et al. (2024), and GITM Zhu et al. (2023), because these methods directly access environment states such as entity coordinates, inventory information, and terrain metadata. In contrast, WISE and all evaluated baselines rely exclusively on raw visual observations. Direct comparison across these settings would confound architectural capability with information privilege and therefore lead to misleading conclusions.

**Evaluation Metrics.** We adopt three complementary evaluation metrics.

- *Map Coverage* measures the fraction of explorable areas visited by the agent and directly evaluates exploration efficiency. Areas that remain unexplored cannot contribute observations for downstream memory or planning.

- *Success Rate* denotes the percentage of episodes that successfully complete all subgoals within a fixed step budget.

- *Average Completion Steps* is computed only over successful episodes and measures task efficiency.

- For sparse sequential (ABA-Sparse) tasks, we additionally report *return localization steps*, defined as the number of steps required to relocate the original location of object A after collecting object B. This metric isolates memory retrieval performance from navigation ability.

**Core hyperparameters.** Key parameters are summarized below, while complete implementation details and selection procedures are provided in Appendix A. For exploration, we set the quadtree depth weight to $\alpha = 0.6$ and distance penalty to $\beta = 0.4$. Frontier utility weights (coverage, novelty, and distance) and Voronoi utility weights (area and distance) are normalized. For memory, the entropy threshold is $\theta_H = 0.15$, cosine similarity threshold is $\theta_c = 0.85$, and top-$k$ retrieval uses $k = 10$. For planning, urgency, semantic relevance, and navigation cost weights are set to $w_u = 0.3$, $w_r = 0.5$, and $w_n = 0.2$, respectively. The maximum quadtree depth is set to 6 for $384 \times 384$ maps and 5 for $128 \times 128$ maps. Voronoi decomposition is updated every 40 steps, with the minimum unexplored area threshold set to 60 pixels.

**VLM cost and latency.** On average, WISE performs approximately 16 GPT-4o calls per episode. Each call consumes roughly 4,575 input tokens and 1,618 output tokens, resulting in an estimated API cost of $0.48 per episode. Since VLM inference runs asynchronously, it does not introduce additional latency into the online control loop.

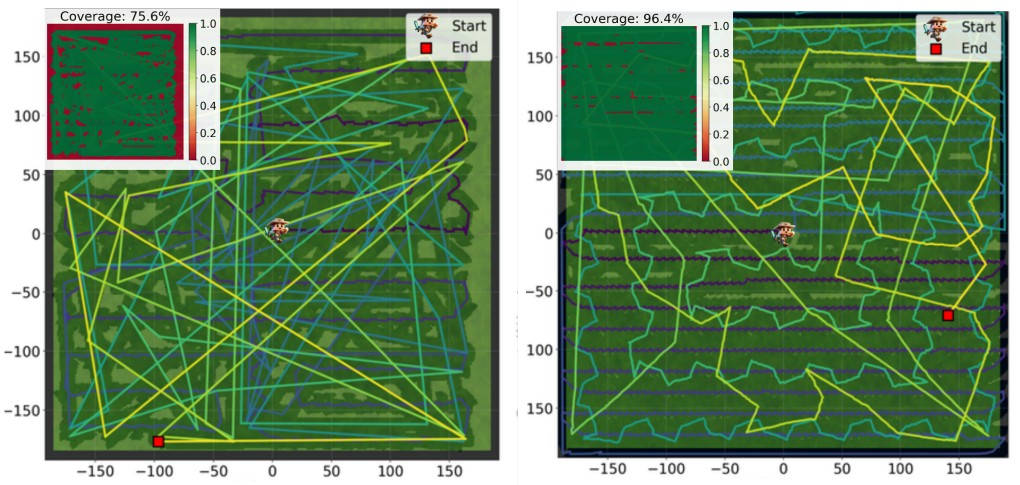

Figure 5: Exploration trajectory comparison on a $384 \times 384$ simulated map after 10,000 timesteps. Left: MrSteve (75.6% coverage). Right: WISE (96.4% coverage). Color indicates temporal progression (blue: early; red: late). MrSteve exhibits locally greedy behavior and repeatedly concentrates exploration near the spawn region, whereas WISE achieves broad and uniform coverage through coordinated global, regional, and local exploration.

## 4.2 Large-Scale Exploration Performance

We first evaluate the efficiency with which each method explores and covers the environment. The primary metric is map coverage, defined as the fraction of unique locations visited by the agent. Table 1 reports coverage under varying step budgets.

**Simulated environment.** We first evaluate under an idealized movement model in which the agent moves deterministically by a fixed distance at each timestep (one block per step). This setting removes low-level execution variability and isolates the intrinsic spatial efficiency of the exploration strategy itself. On the small $128 \times 128$ map with a budget of 6,000 steps, WISE achieves 99% coverage, compared to 59% for MrSteve and 93% for the quadtree-based variant. On the larger $384 \times 384$ map with 20,000 steps, WISE reaches 98% coverage whereas MrSteve plateaus at 83%. The advantage extends beyond final coverage. WISE explores at a rate of 0.097 map coverage per 1,000 steps, compared to 0.067 for MrSteve. Because movement stochasticity has been removed, these results reflect the intrinsic efficiency of the proposed multi-scale exploration strategy.

**Real Environment.** We next evaluate under the standard Minecraft environment containing stochastic elements including mobs, weather, and day–night cycles. Although environmental uncertainty reduces coverage for all methods, WISE consistently maintains a substantial advantage. On the $128 \times 128$ real-world map with a 10,000-step budget, WISE covers 97% of the environment, compared to 67% for MrSteve and only 19% for Steve-1. The three exploration tiers address complementary failure modes of previous approaches. The global quadtree stage performs $O(\log n)$ region selection and avoids local-greedy behavior. The frontier-based regional tier expands exploration boundaries toward unexplored areas. Finally, the Voronoi local completion stage explicitly fills small unvisited regions that remain invisible to higher-level planning. Trajectory visualizations in Figure 5 provide qualitative evidence. MrSteve's count-based exploration tends to concentrate around the spawn location, whereas WISE distributes exploration trajectories more uniformly throughout the environment.

Table 1 further reveals two observations. First, coverage gaps become increasingly pronounced as map scale increases. Although the relative difference narrows from 40 percentage points on small maps to 15 points on large maps, the latter corresponds to substantially larger unexplored regions in absolute area. Second, gains observed in simulation transfer consistently to realistic environments. On the small real-world map,

Table 1: Map Coverage ($\in [0, 1]$) at different timestep budgets. Best results in each column are highlighted in **bold**. WISE* uses global-level search when exploring.

| Method | $128 \times 128$ | | | $384 \times 384$ | | |
|---|---|---|---|---|---|---|
| | 500 steps | 1000 steps | 6000 steps | 1000 steps | 4000 steps | 20000 steps |
| Mrsteve(Count-Based) | 0.46 | 0.57 | 0.59 | 0.17 | 0.53 | 0.83 |
| WISE*(Quadtree-Based) | 0.65 | **0.87** | 0.93 | **0.19** | 0.72 | 0.83 |
| WISE(ours) | **0.66** | 0.85 | **0.99** | **0.19** | **0.73** | **0.98** |

(a) Map Coverage on $128 \times 128$ and $384 \times 384$ map in simulated environment.

| Method | $128 \times 128$ | | $384 \times 384$ | |
|---|---|---|---|---|
| | 5,000 steps | 10,000 steps | 20,000 steps | 40,000 steps |
| Steve-1(random) | 0.11 | 0.19 | 0.06 | 0.13 |
| Mrsteve(count-based) | 0.64 | 0.67 | 0.39 | 0.69 |
| WISE*(Quadtree-Based) | **0.72** | 0.75 | **0.44** | 0.74 |
| WISE(ours) | 0.71 | **0.97** | **0.44** | **0.83** |

(b) Map Coverage of different exploration policies in minecraft.

WISE improves coverage by 30 percentage points over MrSteve and by 78 points over Steve-1, demonstrating strong robustness under environmental stochasticity.

## 4.3 Sequential Sparse Task Performance (ABA-Sparse)

The ABA-Sparse task evaluates long-term memory and spatial recall under sparse-resource settings. The agent must first collect a sparse resource A, then acquire a dense resource B, and finally return to the original location of A without re-exploration. Each episode has a budget of 12,000 timesteps, and success requires completing all three stages. We evaluate each method over 50 episodes on held-out world seeds.

Table 2 summarizes the results. Steve-1 fails completely, achieving a 0% success rate due to the absence of an episodic memory mechanism. MrSteve succeeds in 32% of episodes with an average completion time of 8,123 steps. WISE* (PEM+Graph), which introduces semantic memory but removes multi-scale exploration and opportunistic scheduling, improves success to 47% while maintaining a similar completion time (8,093 steps). The full WISE system achieves a success rate of 62% with an average of only 5,981 steps, corresponding to a 30-point improvement over MrSteve and a 26% reduction in completion time.

The primary challenge arises during the return-to-A phase. MrSteve's Place Event Memory stores raw visual representations and retrieves memories through feature similarity. As shown in Figure 6a Under viewpoint changes, partial occlusion, or environmental variation, cosine similarity frequently drops below the retrieval threshold, causing memory recall failure. Consequently, successful retrieval occurs only in cases where the return trajectory closely matches the original observation viewpoint. WISE addresses this limitation through VLM-generated semantic representations that are substantially more invariant to viewpoint changes. The memory-only variant already improves performance by replacing appearance-based matching with semantic retrieval. The full WISE system further incorporates multi-scale exploration, enabling observations of target locations from diverse viewpoints and producing richer memory representations. This results in an additional 15-point gain over memory-only WISE and reduces average return localization time from 4,253 steps (MrSteve) to 2,341 steps. These results suggest that robust long-term spatial recall in sparse environments requires both semantically grounded memory and sufficiently comprehensive exploration.

## 4.4 Adaptive Non-Sequential Task Performance (ABC-Sparse)

The ABC-Sparse task evaluates opportunistic decision-making under dynamic task conditions. The agent must collect resources A, B, and C, but the completion order is unconstrained. Resource C may appear while the agent is executing another subtask, requiring dynamic reassessment of task priorities. Fixed-sequence planners are expected to systematically miss such opportunities.

Table 2: Performance on Sparse Tasks
ABA sequential sparse task and ABC non-sequential sparse task: success rate and average completion timesteps. WISE(PEM + Graph) removes multi-scale exploration and opportunistic scheduling,. **Bold** = best. — = not applicable.

| Method | Sequential Sparse Task (ABA-Sparse) | | Adaptive Non-Sequential Task (ABC-Sparse) | |
|---|---|---|---|---|
| | Success Rate | Avg. Timesteps | Success Rate | Avg. Timesteps |
| Steve-1 | 0 | — | 0 | — |
| Mrsteve(PEM) | 32 | 8,123 | 33 | 8,040 |
| WISE*(PEM + Graph) | 47 | 8,093 | 42 | 8,325 |
| WISE(ours) | 62 | **5,981** | **77** | **4,620** |

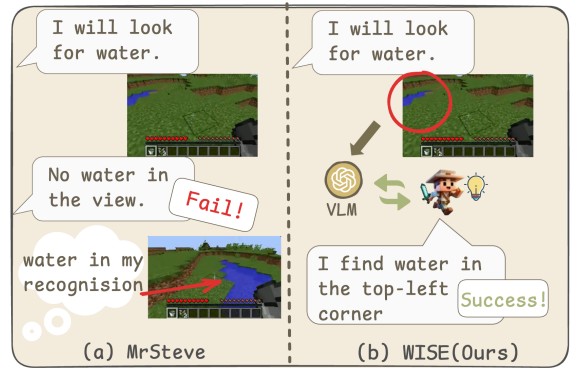 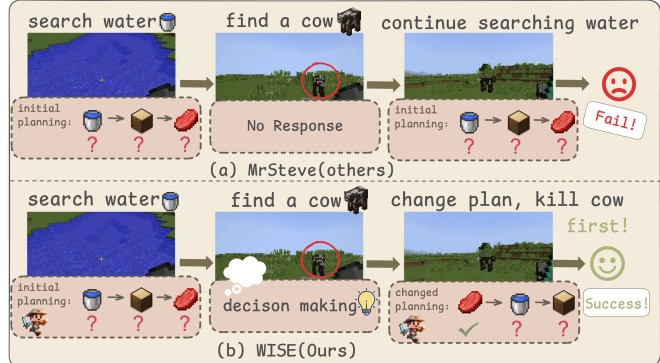

(a) Memory retrieval behaviour in ABA-Sparse Task    (b) Opportunistic task execution in ABC-Sparse Task.

Figure 6: (a) Memory retrieval in the ABA-Sparse task return phase. MrSteve stores raw visual features; under viewpoint changes or occlusion, cosine similarity between current observations and stored features decreases, leading to retrieval failures and forced re-exploration. In contrast, WISE processes keyframes using a VLM that semantically interprets scene content in a viewpoint-invariant manner, storing entities as nodes in a knowledge graph for reliable retrieval. (b) Opportunistic task execution in the ABC-Sparse task. Each row illustrates the agent's planning state over time. Upon detecting resource C (cow), WISE immediately reorders its subtask queue to prioritize the opportunistic encounter, eliminating a future navigation round-trip. MrSteve's fixed-sequence planner continues executing the original plan and misses the opportunity entirely.

Table 2 presents the quantitative results. MrSteve achieves only a 33% success rate with an average completion time of 8,040 steps. Because its planner follows a rigid execution sequence (water → logs → beef), the agent typically finishes water and logs before attempting to locate beef, often requiring a return trip exceeding 150 blocks. In many episodes, cows either cannot be found within the step budget or are encountered too late. WISE* (PEM+Graph), which incorporates semantic memory but lacks opportunistic scheduling, improves success to 42%. Although semantic memory enables later retrieval of cow locations, the fixed task order still prevents immediate exploitation of encountered opportunities. The full WISE system achieves a success rate of 77% with an average completion time of only 4,620 steps, corresponding to a 44-point improvement over MrSteve and a 42.5% reduction in completion steps. The improvement arises directly from the Opportunistic Task Scheduler. When a cow enters the agent's observation space, the Causal Event Graph activates the causal path: $cow \rightarrow CAN\_OBTAIN \rightarrow beef$.

This immediately increases the causal relevance score for the "obtain beef" subgoal while reducing its navigation cost. As shown in Figure 6b, The scheduler recomputes task priorities and dynamically reorders execution to [beef, water, logs]. The agent collects beef immediately and resumes the remaining objectives afterward. This mechanism eliminates expensive future revisitation and directly explains the observed efficiency gain. Failure cases primarily arise when opportunities cannot be exploited in time—for example,

when cows move away or are not detected before leaving the observation range. These results demonstrate that coupling semantic memory with adaptive scheduling transforms the low-level controller from a passive executor into an active decision-maker capable of exploiting environmental opportunities.

## 4.5 Ablation Study

We conduct controlled ablations to quantify the contribution of individual components within WISE. First, we analyze the effectiveness of the exploration hierarchy on a $128 \times 128$ map. Second, we remove each major system component—Causal Event Graph, Multi-Scale Progressive Exploration, and Opportunistic Task Scheduler—to investigate their interactions on the ABC-Sparse task. All ablations use the same hyperparameters and evaluation protocol as the full model.

### 4.5.1 Exploration Tier Ablation

We evaluate three exploration variants on a $128 \times 128$ map: (1) Global-only, which retains only the quadtree-based global exploration tier; (2) Global+Regional, which combines quadtree and frontier-based exploration; and (3) Full WISE, which additionally incorporates the local Voronoi completion stage. Experiments are conducted in the simulated environment to remove low-level movement stochasticity, ensuring that performance differences reflect the intrinsic efficiency of the exploration algorithms.

Table 3b reports map coverage at 5,000, 7,000, and 10,000 timesteps, together with coverage efficiency, defined as coverage per 1,000 timesteps at the 10,000-step budget. The Global-only variant achieves 75% coverage at 10,000 steps, already outperforming MrSteve's count-based strategy (67%). However, coverage rapidly plateaus because large interior regions and boundary-adjacent gaps remain unexplored. Introducing the regional frontier stage increases coverage substantially to 91%, while the complete three-tier design further improves coverage to 97%. Coverage efficiency similarly increases from 0.075 (Global-only) to 0.091 (Global+Regional), reaching 0.097 for the full system. These results indicate that each exploration tier contributes complementary gains.

The improvements can be explained by the distinct roles of each tier. The global quadtree stage prevents the agent from becoming trapped in local neighborhoods by encouraging exploration toward coarse-scale and distant regions. However, once a target region is selected, it does not specify where exploration should continue within that region, often leading to inefficient local wandering. The frontier-based regional stage addresses this limitation by explicitly identifying boundaries between explored and unexplored areas and directing the agent toward informative viewpoints. Nevertheless, even with these two stages, isolated gaps may persist—for example, a small unexplored area hidden behind terrain structures. Such regions frequently remain undetected because they do not lie on active frontiers. The local Voronoi stage explicitly addresses this issue by partitioning the explored space and directing the agent toward the largest remaining unexplored regions. Without this final stage, coverage saturates at 91%; with it, the agent achieves near-complete exploration coverage (97%).

### 4.5.2 Module Ablation

We further evaluate three ablation variants on the ABC-Sparse task using a $128 \times 128$ real-world map. Each variant removes a single component while retaining all others. Specifically, WISE w/o Memory replaces the Causal Event Graph with raw feature-based retrieval; WISE w/o Exploration substitutes the proposed multi-scale exploration strategy with MrSteve's count-based policy; and WISE w/o Opportunistic Task Scheduler adopts a fixed execution order instead of dynamic task prioritization.

**Removing the Causal Event Graph** causes the largest performance degradation, reducing success rate from 77% to 31%, nearly matching MrSteve's 33%. Even with efficient exploration and adaptive scheduling, the absence of semantic memory prevents reliable retrieval of resource locations and eliminates the causal information required for task reasoning. The scheduler itself becomes ineffective because no causal structure exists to support decision-making. This result highlights semantic memory as the foundation of the entire system.

**Removing Multi-Scale Progressive Exploration** reduces success to 45%, corresponding to a 32-point drop relative to the full model. Although semantic memory and opportunistic planning remain available, the agent explores less effectively and consequently acquires fewer useful observations. Resource encounters become increasingly sparse, and observations may be collected from unfavorable viewpoints that fail to generate robust memory representations. These results suggest that exploration quality fundamentally constrains the information available for downstream memory and planning.

**Removing the Opportunistic Task Scheduler** produces a smaller but still significant reduction, decreasing success to 62%. In this setting, the agent can still discover and remember resource locations, but it can no longer dynamically reorder subtasks. As a result, the agent completes water and log collection before revisiting previously observed cows, introducing unnecessary travel costs. The 15-point gap largely corresponds to scenarios in which opportunities are highly time-sensitive.

The impact of removing components follows a clear ordering:

$$\text{Causal Event Graph } (-46) > \text{Multi-Scale Exploration } (-32) > \text{Opportunistic Scheduler } (-15)$$

This hierarchy closely mirrors the dependency structure of WISE itself: exploration supplies observations, memory transforms observations into semantic knowledge, and planning consumes that knowledge to make adaptive decisions. Notably, removing the Causal Event Graph reduces performance to approximately MrSteve's level, demonstrating that efficient exploration alone cannot compensate for missing semantic memory. Conversely, memory without advanced exploration still outperforms MrSteve (45% vs. 33%), indicating that semantic memory retains value even when observation quality is limited. The complete system, integrating all three modules, achieves 77% success and substantially outperforms all two-module variants.

Table 3: Ablation Studies of WISE Method
Ablation results. **Bold** = best. — = not applicable.

| Variant | 5,000 steps | 7,000 steps | 10,000 steps | Cov. Eff. |
|---|---|---|---|---|
| MrSteve (Count-Based) | 0.64 | 0.65 | 0.67 | 0.067 |
| Global only | **0.72** | 0.75 | 0.75 | 0.075 |
| Global + Regional | 0.71 | 0.87 | 0.91 | 0.091 |
| Global + Regional + Local (full WISE) | 0.71 | **0.87** | **0.97** | **0.097** |

(a) Exploration tier ablation on $128 \times 128$ simulated map. Coverage efficiency = coverage per 1,000 timesteps at 10,000-step budget.

| Variant | Success Rate (%) | Avg. Timesteps | $\Delta$ vs. full model |
|---|---|---|---|
| WISE w/o Causal Event Graph | 31 | 8,034 | $-46$ pts |
| WISE w/o Multi-Scale Exploration | 45 | 7,942 | $-32$ pts |
| WISE w/o Opportunistic Task Scheduler | 62 | 6,981 | $-15$ pts |
| WISE (full model) | **77** | **4,620** | — |

(b) Module ablation on adaptive non-sequential (ABC) task ($128 \times 128$ real map). $\Delta$ vs. full model = success rate difference vs. full WISE.

## 5 Conclusion

In this paper, we introduced WISE, a low-level controller for long-horizon embodied tasks in Minecraft that unifies exploration, memory, and planning within a closed reasoning loop. WISE augments conventional episodic memory with a Causal Event Graph that captures not only what, where, and when, but also why past observations matter and which future tasks they enable. Based on this semantic memory, an Opportunistic Task Scheduler dynamically re-prioritizes subtasks, while a multi-scale progressive exploration strategy provides spatially diverse observations. Experiments on sparse sequential and non-sequential tasks

demonstrate that WISE substantially outperforms strong baselines in exploration efficiency, task success rate, and completion speed. Ablation studies further show that the gains arise from the synergy among exploration, semantic memory, and adaptive planning. Future work will extend WISE to broader embodied environments and investigate lightweight alternatives to external VLMs for scalable knowledge construction.

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

# A Implementation Details

## A.1 Multi-Scale Progressive Exploration

The explorer must supply memory with observations that are both spatially complete and efficiently acquired. No single-scale strategy can satisfy both requirements simultaneously: global-only strategies leave local gaps; local-only strategies scale poorly; frontier-only strategies ignore interior voids. WISE addresses this with a **three-tier progressive framework** in which each tier resolves the gaps left by the previous one.

***Global-Scale Rapid Search*** The global tier answers: which macro-region should the agent prioritise next? The Minecraft world is represented as an **adaptive quadtree** whose root spans the full known map and whose children represent candidate regions. Subdivision adapts dynamically to visitation frequency, allocating finer resolution to denser unexplored areas. Each candidate node $N_i$ is scored by:

$$Score(N_i) = \alpha \cdot Depth(N_i) + \beta \cdot Dist(P, N_i) \tag{3}$$

where $Depth(N_i)$ rewards finer-grained unexplored nodes and $Dist(P, N_i)$ penalises navigation cost from the current position P. The quadtree reduces spatial query complexity from $O(n)$ to $O(log n)$, making global planning feasible within Minecraft's $\leq 20$ ms per-step constraint — a threshold that active information-gain methodsChaplot et al. (2020a;b); Zhang et al. (2021) consistently exceed.

***Regional-Scale Frontier Refinement*** Given a macro-region from the global tier, the regional tier answers: which boundary points offer the highest information gain? Frontier points — boundaries between explored and unexplored space — are detected via morphological dilation and scored as:

$$Score(f_i) = \alpha \cdot Cover(f_i) + \beta \cdot Nove(f_i) + \gamma \cdot Dist(f_i) \tag{4}$$

where $Cover(f_i)$ is uncovered area in the visible range, $Nove(f_i)$ is information-entropy gain, and $Dist(f_i)$ is distance from historical viewpoints. The three terms jointly reward novelty, penalise revisitation, and encourage spatial spread — countering the local-optimum tendency of count-based methods that optimise only the first term.

***Local-Scale Voronoi Completion*** After frontier refinement, residual interior patches remain — a problem that neither global planning nor frontier-following resolves. The local tier answers: which residual gaps remain unfilled? We apply Voronoi decompositionKrozel & Andrisani (1990) over the visited point set $\mathbf{V} = \{v_1, v_2, \ldots, v_n\}$ (where $v_i = (x_i, y_i, t_i)$, $x_i$ and $y_i$ are 2D coordinates, $t_i$ is the timestamp) to identify uncovered regions $\mathbf{U} = \mathbf{\Omega} \setminus \bigcup_{i=1}^{n} \mathbf{C_i}$, where:

$$\mathbf{C_i} = \{p \in \mathbf{\Omega} \mid \|p - v_i\|_2 < \|p - v_j\|_2, \ \forall j \neq i\} \tag{5}$$

Each residual region $U_i$ is prioritised by:

$$Score(U_i) = \alpha \cdot Area(U_i) + \beta \cdot Dist(P, U_i) \tag{6}$$

This provides a *formal local completeness guarantee*: every point in $\omega$ will eventually be assigned to a Voronoi cell and visited. Together, the three tiers constitute the first exploration strategy that is simultaneously globally efficient, boundary-aware, and locally complete.

***Local escape mechanism*** Three escalating recovery heuristics prevent exploration stagnation. If fewer than five blocks are traversed in 30 seconds, the agent triggers *local replanning*. On repeated failure to reach a target, it performs a *target reset* via memory query. On repeated location revisitation, it executes a *graph-level rollback* to the nearest unvisited key node. These heuristics ensure that exploration continues to supply fresh observations to the memory module even under adverse terrain conditions.

## A.2 Hardware and Software Configuration

All experiments were conducted on NVIDIA RTX4080 16GB. The host operating system was Ubuntu v22.04. All agents were implemented in Python v3.11 with PyTorch v2.5.5. Minecraft Java Edition was accessed via MineDoJo v0.1 (built on MineRL). The VPT base controller checkpoint used was VPT-Nav(MrSteve). GPT-4o API calls were made via OpenAI API version gpt-4o-2024-11-20. Each experiment episode ran with a per-step time constraint of $\leq 20$ ms, consistent with the Minecraft Java Edition real-time rendering requirement.

## A.3 World and Map Configuration

**Sparse task experiments (Section 4.2).** A $128 \times 128$ real Minecraft map combining four biomes was used as shown as Figure 7. Each method was evaluated over 50 episodes on world seeds.

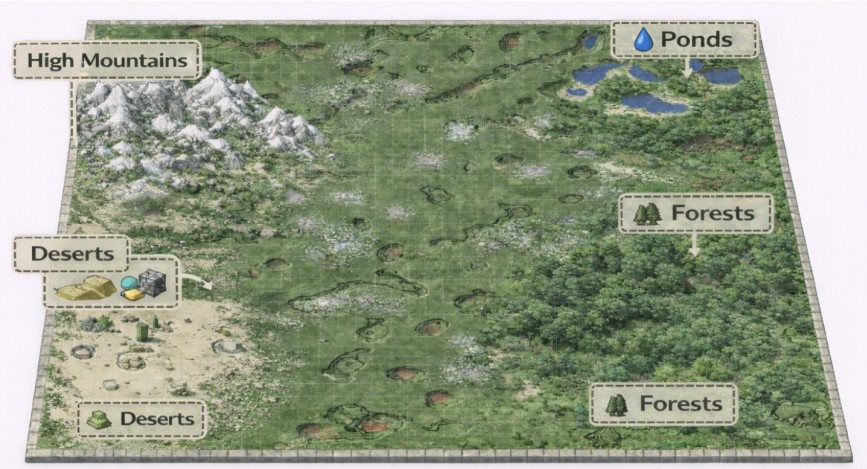

Figure 7: The 128×128 real Minecraft map used for all task-completion experiments. The combination of biome diversity and deliberate resource sparsity requires all three of WISE's modules to operate in concert.

**Exploration experiments (Section 4.3).** Two map scales were used: small ($128 \times 128$ blocks) and large ($384 \times 384$ blocks), each evaluated under two conditions.

- **Simulated**: mobs disabled, terrain variation disabled, day–night cycle paused.
- **Real (Stochastic)**: full Minecraft stochasticity including hostile mobs, weather, and day–night cycles.

All agents started from the same fixed spawn point on the same world seed within each condition. World seeds used: random from seed = 42 (small), to seed = 103 (large).

## A.4 Baseline Implementation

**Steve-1**: the publicly released checkpoint was used without modification; it serves as a no-memory reference baseline. **MrSteve**: the publicly released checkpoint and Place–Event Memory (PEM) implementation were used; it serves as the primary baseline. All baselines share identical map configurations, spawn points, and episode budgets with WISE.

## A.5 Full Hyperparameter Table

Table 4 extends Table 1 in the main paper with all secondary parameters required for reproduction.

Table 4: Full hyperparameter list for WISE.

| Module | Parameter | Value | Description |
|---|---|---|---|
| Exploration | Quadtree depth weight $\alpha$ | 0.6 | Rewards unexplored depth (Eq. 3) |
| | Quadtree distance penalty $\beta$ | 0.4 | Penalises navigation cost (Eq. 3) |
| | Frontier weights $(\alpha, \beta, \gamma)$ | norm. | Coverage, novelty, distance (Eq. 4) |
| | Voronoi weights $(\alpha, \beta)$ | norm. | Area, distance (Eq. 5) |
| | Stagnation threshold | 5 blk/30 s | Triggers local replanning |
| | Sliding window size | 60s | Short-term memory eviction |
| Memory | Entropy threshold $\theta_H$ | 0.15 | Entropy-Based keyframe trigger |
| | Cosine threshold $\theta_c$ | 0.85 | Cluster-Based trigger |
| | Grounding threshold $\theta_{grid}$ | 0.9 | KG entry matching |
| | Top-$k$ retrieval | 10 | Short-term candidates |
| | Re-ranking weight $\lambda$ | 0.5 | Similarity vs. recency |
| | VLM calls per episode | 16 | Average across episodes |
| Planning | Urgency weight $w_u$ | 0.3 | (Eq. 2) |
| | CausalRel weight $w_r$ | 0.5 | (Eq. 2) |
| | Navigation cost weight $w_n$ | 0.2 | (Eq. 2) |

**Hyperparameter selection.** Keyframe thresholds ($\theta_H$, $\theta_c$) were selected via grid search on a held-out validation map. Search ranges: [ $\theta_H \in \{0.05, 0.10, 0.15, 0.20\}$, $\theta_c \in \{0.75, 0.80, 0.85, 0.90\}$].

## A.6 Knowledge Graph Construction Details

The structured Minecraft knowledge database used for grounding GPT-4o descriptions was derived from Wikipedia and covers mob drop relations (e.g., cow $\rightarrow$ drops $\rightarrow$ beef), crafting recipes, smelting chains, and environmental state transitions. Total number of database entries: 1534

Graph schema:

- **Node types:** entity (e.g., cow, stone), action (e.g., mine, craft), environment (e.g., rain, hunger).

- **Edge types:** timestep, position, causal.

Average number of nodes and edges per episode at termination:120.

## A.7 API Cost Breakdown

All GPT-4o calls were processed *asynchronously* and did not contribute to the per-step latency budget as follows in Table 5

Table 5: GPT-4o API cost breakdown.

| Item | Value |
|---|---|
| Avg. VLM calls per episode | ~16 |
| Avg. input tokens per call | 4575 |
| Avg. output tokens per call | 1618 |
| Est. cost per episode (USD) | 0.48$ |
| Total API cost across all exps. (USD) | 512$ |

# B  Task Definitions

## B.1  ABA-Sparse (Sequential Task)

The agent spawns in the $128 \times 128$ real Minecraft map. The task proceeds in three phases:

1. **Phase $A_1$**: Locate and collect resource $A$ at position $p_A$.

2. **Phase $B$**: Navigate to and collect distant resource $B$ (minimum distance from $A$: $\geq 100$ blocks.

3. **Phase $A_2$**: Return precisely to position $p_A$ using only memory — re-exploration is prohibited as the remaining timestep budget is insufficient.

A trial is **successful** if all three items are collected within the timestep budget. Timestep budget: 12000.

## B.2  ABC-Sparse (Non-Sequential Task)

The agent must collect items $A$, $B$, and $C$ in *any* order. Resource $C$ (e.g., beef from a cow) appears *opportunistically* mid-episode at a random location during execution of sub-task $A$, requiring immediate priority reassessment. Any fixed-sequence policy systematically misses $C$ and incurs a full round-trip penalty.

Specific resource types:    $A = $ **water,** $B = $ **logs,** $C = $ **beef (from cow)**]. Timestep budget: **12000**. $C$ spawn distribution: uniform random within 30 blocks of the agent at step 500.

