# OpenReview forum: "WISE: A Long-Horizon Agent in Minecraft with Why-Which Reasoning"
_TMLR — Under review for TMLR_

### Review · Reviewer_93fx · 2026-06-23

**Summary Of Contributions:**

The paper proposes a long-horizon agent framework named WISE with three key designs: causal episodic memory, opportunistic task scheduling, and multi-scale progressive exploration. The core lies in the construction of VLM-driven causal graph and uses it for causal augmentation during memory retrieval. The opportunistic task scheduler is designed to reprioritize subtasks based on causal relevance. Finally, a three-tier exploration strategy is presented to improve coverage efficiency in a multi-scale manner. Experimental studies show that WISE achieves superior performance to baselines in terms of exploration coverage, task success rate, as well as completion time. Ablation studies were performed to demonstrate the effectiveness of each design component.

Strengths:
1. Clear motivation: existing methods didn’t consider causal relationships at low-level control.

2. Simple design: the technical design is simple and easy to understand.

3. Experiments: the proposed WISE achieves impressive performance compared to baselines in various aspects.

Weaknesses:

1. Many technical details are lacking in the main text.

2. Notations are messy and needs substantial improvement.

3. Experimental studies are insufficient.

4. It would be nice to extend the proposed method to a wider context rather than limited for Minecraft.

**Additional Comments:**

Nil

**Audience:**

Yes

**Audience Explanation:**

The proposed framework is practical and interesting, especially for the decoupling of the episodic memory from which-why reasoning.

**Broader Impact Concerns:**

Though the submission does not include a Broader Impact Statement, there are no major broader-impact concerns.

**Claims And Evidence:**

Yes

**Claims Explanation:**

The three key capabilities of WISE claimed in the paper are supported by experimental studies.

**Requested Changes:**

1. Please provide more details in the technical part.
- Explain Algorithm 1 in greater detail.
- When introducing O_t, though understandable, the notations H, W, and C are not defined.
- How does the VLM inference be executed independently? How to keep the causal graph up to date for causal use?
- How many keyframes are obtained?
- Urgency, NavCost, and CausalRel in Eq. (2) are not defined. Only providing their intuitions is not sufficient.
- Various utilities in Section 3.3 are not defined. They appear to be in the Appendix. Please move them to the main text.

2. Notations are messy and needs substantial improvement.
- Notation presentation in Algorithm 1 is messy. Please keep them consistent with those in the main text.
- Avoid using the same notation for different purposes. For instance, H is used for both height and window size; alpha and beta are used in different equations as well.
- The e_x and Enct in Eq. (1) are not defined.

3. Experimental studies need to be enhanced.
- As the causal graph is the core design, it is essential to evaluate how accurate the VLM inference and the causal matching are.
- How often does the reprioritizing happen in the tested experiments?
- There are many hyperparameters in WISE. Their sensitivity should be tested. It is insufficient to fix most of them to be a single value and report the results.
- In Section 4.3, it is stated that the failure of MrSteve is because cosine similarity frequently drops below the retrieval threshold. How about the performance of the models when tuning this threshold? Will the performance gap mainly due to this threshold set to be too high?
- The experiments were conducted on short sequences, which doesn’t mimic the real scenarios well. It would be desirable to be tested on long sequences with a mixture of different types of sparse tasks.
- When moving from simulated environment to real one in Section 4.2, why the performance of MrSteve improves but that of WISE drops under comparable setup?
- In Section 4.4, did it test only a single task type, i.e., “water→logs→beef”? If yes, it is too limited and the results may not be that convincing.
- In Table 3: why the performance of “Global + Regional” and “full WISE” is worse than that of “Global only” at 5,000 steps? This is unexpected and needs to be discussed.


4. It would be nice to extend the proposed method to a wider context rather than limited for Minecraft. Can WISE be used in other contexts? If yes, what modifications are needed?

---

> ### Author Response · Authors · 2026-07-16
> **Rebuttal to Reviewer  93fx Part 1**
>
> anuscript required clearer technical details, more consistent notation, and stronger experimental validation. We have revised the manuscript accordingly and address each concern below.
>
> **Concern1: Please provide more details in the technical part.**
>
> **Concern1.1: Algorithm 1 and implementation details**
>
> **Response:**
>
>  We agree that the original Algorithm 1 was overly compressed. In the revision, we rewrite it as a clearer single-decision loop and explicitly define all variables. The revised algorithm separates four stages: memory update, semantic affordance retrieval, task prioritization, and low-level execution.
>
> We further clarify that VLM inference is executed asynchronously rather than inside the real-time control loop. Selected keyframes are queued for VLM processing, while the agent continues acting using short-term memory and the latest committed SAEG. Once VLM outputs are available, SAEG is updated atomically with grounded entities, spatial attributes, timestamps, and semantic affordance relations. This design avoids blocking execution while keeping the semantic memory continuously updated. To improve reproducibility, we also add formal definitions of the retrieval score, affordance matching term, scheduler priority score, urgency, navigation cost, and affordance relevance in the revised manuscript. For readability, we do not reproduce all equations in the response letter, but they are now included in the revised method section together with the updated pseudocode.
>
> We also clarify the keyframe processing strategy. The VLM worker processes keyframes in groups of 16 frames or flushes the queue after 25 seconds to balance latency and efficiency. For a typical $128\times128$ task, approximately 200--300 keyframes are processed per episode.
>
> **Concern1.2:  Notation clarification**
>
> **Response:**
>
> We revise the observation definition and notation throughout the manuscript. Specifically, image dimensions are explicitly defined as $H_{\mathrm{img}}, W_{\mathrm{img}}, C_{\mathrm{img}}$, and we avoid reusing symbols for different purposes. We also define all terms in the retrieval and scheduling equations, including $e_x$, $z_\tau$, $Enc_{\mathrm{vis}}$, and $Enc_{\mathrm{text}}$.
>
> The retrieval score is revised as:
>
> $$
> Score(x,\tau)=\lambda \cdot sim(e_x,z_\tau)+(1-\lambda)\cdot AffordanceMatch(x,\tau),
> $$
>
> where $AffordanceMatch$ measures whether a memory candidate contains an entity connected to the task through SAEG.
>
> Similarly, the scheduler priority is explicitly defined as:
>
> $$
> Priority(\tau_i,t)=w_u Urgency(\tau_i,t)+w_a AffordanceRel(\tau_i,t)+w_n[1-NavCost(\tau_i,t)].
> $$
>
> We move the global quadtree utility, regional frontier utility, and local Voronoi completion utility from the appendix to Section 3.3.
>
> **Concern2 : Notations need substantial improvement.**
>
> **Response:**
>
> We agree and perform a complete notation cleanup. A notation table is added, and symbols are unified across the main text, algorithms, equations, and appendix. For example, $H_{\mathrm{img}},W_{\mathrm{img}},C_{\mathrm{img}}$ denote image dimensions, $T_{\mathrm{win}}$ denotes the MineCLIP temporal window, $e_x$ denotes memory visual embeddings, and $z_\tau$ denotes task embeddings.

---

> > ### Author Response · Authors · 2026-07-16
> > **Rebuttal to Reviewer 93fx Part 2**
> >
> > **Concern3:  Experimental studies need to be enhanced.**
> >
> > **Concern3.1: Evaluation of VLM inference and SAEG accuracy**
> >
> > **Response:**
> >
> > We agree that SAEG quality should be evaluated explicitly. We add an analysis using 50 sampled keyframes from ABC experiments with manual annotations of entities and semantic affordance relations. We report entity recognition precision and affordance relation matching precision against human annotations and a curated Minecraft knowledge base.
> >
> > |      | entity precision | affordance relation matching precision |
> > | ---- | ---------------- | -------------------------------------- |
> > | SAEG | 87.5%            | 87.5%                                  |
> >
> > **Concern3.2: How often does the reprioritizing happen in the tested experiments?**
> >
> > **Response:**
> >
> > Task reprioritization occurs when the agent discovers resources relevant to future subtasks during execution. For example, observing a cow activates the relation `cow → CAN_OBTAIN → beef`, allowing the scheduler to move “obtain beef” earlier. We clarify this behavior and provide additional analysis in the revision.
> >
> > **Concern3.3: Hyperparameter sensitivity**
> > **Response:**
> >
> >
> >
> > We add sensitivity studies for key parameters affecting memory construction, exploration, and scheduling, including VLM keyframe batch size, keyframe extraction thresholds, quadtree parameters, and scheduler weights. The results show that the default configuration provides a good trade-off between semantic update frequency, computational cost, exploration efficiency, and task completion.
> >
> > | VLM batch size  per request           | 12   | 16    | 20    |
> > | ------------------------------------- | ---- | ----- | ----- |
> > | Average response time per Keyframe(s) | 39s  | 44.1s | 51.1s |
> > | Average VLM requests per episode      | 19   | 14    | 11    |
> >
> >
> >
> > The results show the trade-off between request frequency and response latency: smaller batches lead to more VLM requests, while larger batches reduce the number of requests but increase the average response time per keyframe. The default setting provides a balanced choice between graph update frequency and VLM efficiency.
> >
> > **Retrieval threshold of MrSteve**
> >
> > | ($ \theta_c, \theta_H$)  | (0.65,0.35) | (0.75,0.25) | (0.85,0.15) | (0.95,0.05) |
> > | ------------------------ | ----------- | ----------- | ----------- | ----------- |
> > | Total selected keyframes | 357         | 281         | 217         | 425         |
> >
> >
> >
> > Lowering the retrieval threshold improves recall but introduces irrelevant memories, while increasing it improves precision but misses relevant observations under viewpoint changes and occlusion. Therefore, the performance gap is not caused by an inappropriate threshold alone, but by the limitation of purely visual similarity retrieval for task-level relevance.
> >
> >  (3) the quadtree depth parameter used in global exploration.
> >
> > | Quadtree depth weight | 0.48 | 0.6  | 0.72 |
> > | --------------------- | ---- | ---- | ---- |
> > | 5000 timesteps        | 0.56 | 0.71 | 0.52 |
> > | 10000 timesteps       | 0.86 | 0.97 | 0.84 |
> >
> >
> >
> >  (4) the scheduler priority weights (w_u,w_a,w_n), where waw_awa denotes the semantic affordance relevance weight.
> >
> > | $(w_u,w_a,w_n)$ | （0.3，0.5，0.2） | （0.5，0.3，0.2） | （0.3，0.2，0.5） | （0.3，0.4，0.3） |
> > | --------------- | ----------------- | ----------------- | ----------------- | ----------------- |
> > | Success Rate    | 77%               | 68%               | 56%               | 74%               |
> > | Total steps     | 4620              | 5830              | 6650              | 5225              |
> >
> > The results confirm that affordance-aware scheduling improves task efficiency, while extreme weighting choices may lead to less balanced exploration-exploitation behaviors.
> >
> > **Concern 3.4:  In Section 4.3, it is stated that the failure of MrSteve is because cosine similarity frequently drops below the retrieval threshold. How about the performance of the models when tuning this threshold? Will the performance gap mainly due to this threshold set to be too high?**
> >
> > **Response:**
> >
> > Lowering the threshold can indeed improve recall, but it also introduces many irrelevant memories. A lower threshold tends to retrieve scenes that are visually similar but unrelated to the current task, causing the agent to navigate to incorrect locations and waste steps. Conversely, increasing the threshold may improve precision, but it can also cause the agent to miss relevant memories under viewpoint changes, occlusion, or lighting variations. Therefore, the issue is not simply that the threshold is “too high”; rather, retrieval based solely on visual similarity cannot reliably capture task-level semantic relevance.

---

> > > ### Author Response · Authors · 2026-07-16
> > > **Rebuttal to Reviewer 93fx Part 3**
> > >
> > > **Concern3.5: The experiments were conducted on short sequences, which doesn’t mimic the real scenarios well. It would be desirable to be tested on long sequences with a mixture of different types of sparse tasks.In Section 4.4, did it test only a single task type, i.e., “water→logs→beef”? If yes, it is too limited and the results may not be that convincing.**
> > >
> > > We thank the reviewer for this valuable suggestion. We agree that the original sparse-task evaluation was limited in task diversity and sequence length. To address this concern, we add a new **Mixed-5 Sparse Task** experiment containing five heterogeneous objectives:
> > >
> > > [obtain beef→obtain wheat seeds→collect logs→obtain wool→find water].
> > >
> > > This setting includes diverse objectives involving mobs, vegetation, collectible resources, and environmental landmarks, requiring longer-horizon memory retrieval, semantic affordance reasoning, and dynamic task prioritization.
> > >
> > > | Method       | Success Rate |
> > > | ------------ | ------------ |
> > > | Steve-1      | 0%           |
> > > | MrSteve(PEM) | 16%          |
> > > | WISE         | 56%          |
> > >
> > > The results show that WISE maintains its advantage on longer and more diverse sparse-task sequences. We include this experiment in the revised manuscript and clarify the difference between symbolic task length and embodied execution horizon.
> > >
> > > **Concern3.6: When moving from simulated environment to real one in Section 4.2, why the performance of MrSteve improves but that of WISE drops under comparable setup?**
> > >
> > > We thank the reviewer for pointing out this confusing comparison. The simulated environment uses an idealized deterministic motion model, whereas the real Minecraft environment introduces control noise, terrain variation, weather, mobs, and day-night changes. Therefore, the same timestep budget may lead to different movement and coverage behaviors.
> > > The apparent improvement of MrSteve in the real setting is partly due to the coverage metric: counting visited locations can favor stochastic trajectories that spread over more cells, especially on smaller maps. This does not necessarily indicate more effective exploration. In contrast, WISE relies on structured exploration targets from the quadtree, frontier, and local completion modules, whose early-stage performance can be affected by low-level navigation uncertainty.
> > > With larger timestep budgets, WISE recovers its advantage and achieves higher final coverage due to its structured exploration strategy. We have clarified this distinction in the revised discussion of Section 4.2/Table 1.
> > >
> > > **Concern3.7: In Table 3: why the performance of “Global + Regional” and “full WISE” is worse than that of “Global only” at 5,000 steps? This is unexpected and needs to be discussed.**
> > >
> > >
> > >
> > > **Response:**
> > >
> > > We thank the reviewer for pointing out this observation. The difference at 5,000 steps is marginal: Global-only achieves 0.72 coverage, while Global+Regional and full WISE achieve 0.71, with only a 0.01 gap. We therefore consider it an early-stage variation rather than a meaningful performance degradation.
> > >
> > > This occurs because Global-only prioritizes rapid coarse-scale expansion, whereas the regional tiers allocate part of the early budget to frontier refinement. Low-level control uncertainty and navigation recovery in the Minecraft environment may further introduce small fluctuations. As the exploration horizon increases, the benefits of regional refinement become evident, with Global+Regional and full WISE outperforming Global-only at 7,000 and 10,000 steps. We have added this clarification to the revised manuscript.

---

> > > > ### Author Response · Authors · 2026-07-16
> > > > **Rebuttal to Reviewer 93fx Part 4**
> > > >
> > > > **Concern4: It would be nice to extend the proposed method to a wider context rather than limited for Minecraft. Can WISE be used in other contexts? If yes, what modifications are needed?**
> > > >
> > > > **Response:**
> > > >
> > > > We thank the reviewer for this suggestion. We agree that the current evaluation is limited to Minecraft, and we have added a discussion on the broader applicability of WISE in the revised manuscript.
> > > >
> > > > WISE is not fundamentally tied to Minecraft. Its core components—semantic affordance memory, exploration, and opportunistic task scheduling—can be transferred to other embodied or interactive environments with visual observations, spatial information, and callable low-level skills. Adapting WISE mainly requires replacing three components: the low-level controller, the semantic affordance graph schema, and the memory representation. For example, Minecraft relations such as `cow → beef` can be replaced by domain-specific relations such as `fridge → contains → food` in household environments or `button → opens → page` in GUI tasks. Similarly, 3D coordinates can be replaced by SLAM maps, topological scene graphs, DOM structures, or transition graphs. We have added this discussion to the limitation section and clarified the potential extensions of WISE beyond Minecraft.

---

> > > > > ### Comment · Action_Editor_A2UX · 2026-07-16
> > > > >
> > > > > Dear Reviewer,
> > > > >
> > > > > The authors have now submitted their rebuttal and revised manuscript. Could you please review the revised submission and the authors' responses, and provide any further feedback or final recommendations at your earliest convenience?
> > > > >
> > > > > Thank you for your contribution to TMLR
> > > > >
> > > > > Best, AE

---

### Review · Reviewer_g3W8 · 2026-06-30

**Summary Of Contributions:**

The paper proposes WISE, a Minecraft agent framework for long-horizon sparse tasks. The main contributions are a Causal Event Graph for semantic memory retrieval, an Opportunistic Task Scheduler for dynamically reordering subtasks, and a multi-scale exploration strategy for improving environment coverage. The paper shows improved success rates and efficiency over Steve-1 and MrSteve on sequential and non-sequential sparse Minecraft tasks.

The main strengths are that the paper addresses an important bottleneck in long-horizon embodied agents, combines memory, exploration, and scheduling in a unified framework, and provides ablations for the proposed components. The main weaknesses are that the implementation details of several modules are under-specified, the causal-reasoning claim appears overstated, and the experiments may confound causal reasoning with perception, semantic retrieval, and exploration quality.

**Audience:**

Yes

**Audience Explanation:**

The paper addresses an important problem in long-horizon embodied agents: improving memory, exploration, and adaptive task execution in sparse Minecraft environments.

**Broader Impact Concerns:**

I do not identify any major broader impact concerns that require additional discussion.

**Claims And Evidence:**

No

**Claims Explanation:**

Partially supported.

The experimental results support the claim that ***WISE improves task success and efficiency over the selected baselines on the proposed Minecraft tasks.*** However, the claims about causal reasoning are not fully supported by evidence. The proposed Causal Event Graph appears to rely on semantic/world knowledge provided by a VLM, rather than causal discovery through interaction. Relations such as `cow → beef` or `tree → logs` are better interpreted as semantic affordance links or object–resource associations.

**Requested Changes:**

### 1. Clarity on how the method is implemented

- This is currently the most serious concern as it makes justification of the paper very difficult. The implementation details of the proposed modules are not clear at all. Several key components remain under-specified, including the exact VLM prompting and filtering procedure for constructing the Causal Event Graph.

- The use of the scoring functions is also unclear. For example, when a retrieved memory is selected, what threshold triggers navigation to a memory location, how ties or conflicting scores are handled, and how the memory score interacts with the scheduler’s priority score. Clarifying the role of these scores would make the method easier to understand and reproduce. Variables in every equation should be presented in pseudocode to provide a high-level explanation. It's even better if these variables can also be presented in a diagram.

- The VLM update procedure is unclear. The paper states that memory is updated continuously, but also reports only around 16 GPT-4o calls per episode. This is confusing.

- It is unclear how memory is integrated with the executor. The paper appears to use memory to retrieve task-relevant locations and then call VPT-Nav/Steve-1 to navigate and execute, but memory does not seem to be directly provided to the low-level policy.

- Moreover, the VLMs used in WISE and MrSteve are not clearly specified. It is possible that WISE outperforms the baselines partly because GPT-4o has stronger semantic reasoning capabilities, rather than because of the proposed causal-memory mechanism.


***I strongly suggest rewriting the method section. The main criterion should be that readers can understand how to re-implement the method without access to the authors’ code.***


---

### 2. Causal-reasoning overclaim

The paper may make overly strong claims about causal reasoning, but the proposed Causal Event Graph appears to rely largely on knowledge encoded in the language data used to train the VLM. True causal reasoning should involve causal discovery through interaction. Relations such as `cow → beef` or `tree → logs` are better described as semantic affordance links or object–resource associations.

The paper should clarify that the method performs VLM-assisted semantic retrieval and opportunistic scheduling, and should tone down or remove the claim about causal reasoning. If the paper aims to claim causal reasoning, stronger evidence is needed, such as learning novel environment-specific cause-and-effect relations from interventions.

---

### 3. The claim that the proposed tasks require causal reasoning is not fully convincing

Under perfect state estimation or symbolic abstraction, relations such as `cow → beef` are simple object–resource affordances that a strong RL method or planner could plausibly learn without an explicit causal graph. This is supported by benchmarks such as Craftax [1], where non-LLM RL agents operate over symbolic observations and can learn substantially more complex survival and crafting behaviours.

Therefore, the difficulty of the proposed Minecraft tasks may be confounded with perception, state estimation, and memory retrieval under pixel observations, rather than isolating causal reasoning. The paper would be stronger if it included experiments with oracle symbolic state, controlled perception, or non-LLM RL/planning baselines to show that the Causal Event Graph provides benefits beyond semantic abstraction and retrieval.

---

### 4. More ablation studies and analysis are required

The proposed tasks may be solvable by simply providing explicit instructions in the VLM prompt, such as “beef comes from cows.” Combining such predefined instructions with a sufficiently strong exploration strategy could potentially achieve high performance. Therefore, the proposed Causal Event Graph may not outperform a simpler instruction-based baseline. The paper should include additional ablations comparing the Causal Event Graph against such simple prompting or semantic-retrieval baselines.

---

### Minor comment

In Figure 6(a), “recognision” should be corrected to “recognition.”

---

### Reference

[1] Craftax: A Lightning-Fast Benchmark for Open-Ended Reinforcement Learning

---

> ### Author Response · Authors · 2026-07-16
> **Rebuttal to Reviewer g3W8 Part 1**
>
> We thank the reviewer for the careful and constructive feedback. We appreciate the recognition that WISE addresses an important bottleneck in long-horizon embodied agents and improves task success and efficiency over evaluated baselines.
>
> We have revised the manuscript accordingly and respond to each concern below.
>
> **Concern 1: Clarity on implementation details**
>
> **Response:**
> We agree that the original manuscript lacked sufficient implementation details for reproducibility. We have substantially revised Section 3 and Appendix A to clarify the complete pipeline.
>
> Specifically, we provide the VLM prompting procedure for SAEG construction, including input format, output schema, node/edge definitions, entity normalization, duplicate merging, and spatial-temporal association. We also clarify that graph edges represent **semantic affordance relations**, rather than causal relations. For example, `cow → CAN_OBTAIN → beef` indicates that a cow enables obtaining beef under Minecraft mechanics and is used for retrieval and scheduling.
>
> We further clarify the graph update mechanism. WISE does not invoke the VLM at every environment step. Instead, observations are filtered using clustering- and entropy-based keyframe selection, and only informative keyframes are asynchronously processed by the VLM. This explains why WISE uses approximately 16 GPT-4o calls per episode rather than one call per timestep.
>
> In addition, we revise the scoring functions and pseudocode by replacing causal terminology with affordance-based terminology. The memory retrieval score is updated as:
>
> $$
> Score(x,\tau_n)=\lambda\cdot sim(e_x,Enc_t(\tau_n))+(1-\lambda)\cdot AffordanceMatch(x,\tau_n),
> $$
>
> where $AffordanceMatch$ measures whether an observed entity is connected to a task through SAEG. Similarly, the task scheduler uses:
>
> $$
> Priority(\tau_i,t)=w_u\cdot Urgency(\tau_i,t)+w_r\cdot AffordanceRel(\tau_i,t)+w_n\cdot [1-NavCost(\tau_i,t)].
> $$
>
> We explicitly distinguish the roles of the two modules: retrieval identifies relevant memory locations for a subgoal, while the scheduler determines which pending subgoal should be executed next. SAEG is not directly provided to the low-level controller; instead, it influences navigation targets and task ordering, while VPT-Nav and Steve-1 execute the selected goals.
>
> Finally, we clarify VLM usage across methods. WISE uses GPT-4o for SAEG construction, whereas MrSteve does not use GPT-4o-based semantic graph construction. We also ensure that the GPT-4o model identifier is reported consistently throughout the manuscript.

---

> > ### Author Response · Authors · 2026-07-16
> > **Rebuttal to Reviewer g3W8 Part 2**
> >
> > **Concern 2: causal- reasoning overclaim**
> >
> > **Response:**
> > We agree that the previous causal terminology was stronger than necessary. In the revision, we consistently replace “causal reasoning” with **semantic affordance reasoning** and rename the graph as SAEG.
> >
> > We revise the abstract, introduction, method, captions, and discussion to clarify that WISE improves long-horizon embodied control by connecting episodic observations with semantic affordances, enabling more robust memory retrieval and opportunistic task scheduling under sparse observations.
> >
> > **Concern 3: The claim that the proposed tasks require causal reasoning is not fully convincing**
> >
> > **Response:**
> >
> > We agree with the reviewer’s point and revise the task description accordingly. Rather than claiming that ABA-Sparse and ABC-Sparse require formal causal reasoning, we describe them as sparse long-horizon visual-control tasks requiring semantic memory retrieval and adaptive execution.
> >
> > The challenges mainly arise from sparse entity discovery, memory construction, viewpoint-robust retrieval, navigation back to relevant locations, and dynamic subtask ordering. SAEG helps the agent identify why an observation is relevant to a future task and prioritize appropriate actions, which is better characterized as affordance-grounded task reasoning.
> >
> > We also add a limitation discussion noting that the current evaluation does not fully disentangle semantic reasoning from perception and exploration quality, and that controlled oracle-state or planning-based comparisons would provide further insight.
> >
> > **Concern 4: More ablation studies and analysis are required**
> >
> > **Response:**
> >
> > We add a new ablation baseline, **VLM Entity Retrieval without SAEG**, which uses GPT-4o-generated entity labels for retrieval but does not construct affordance edges or perform graph traversal. This experiment isolates the contribution of SAEG beyond VLM-based semantic recognition.
> >
> > | Method                  | Success Rate | Avg. TimeSteps |
> > | ----------------------- | ------------ | -------------- |
> > | WISE w/o graph + prompt | 56%          | 6320           |
> > | WISE    (full model)    | 77%          | 4620           |
> >
> > The results confirm that VLM labels help semantic recognition, but SAEG contributes structured semantic affordance linking, reusable affordance memory, and more stable task-level retrieval.
> >
> > **Minor comment**
> >
> > **Response:**
> > We thank the reviewer for identifying this typo. We correct “recognision” to “recognition” in Figure 6(a).

---

> > > ### Comment · Action_Editor_A2UX · 2026-07-16
> > >
> > > Dear Reviewer,
> > >
> > > The authors have now submitted their rebuttal and revised manuscript. Could you please review the revised submission and the authors' responses, and provide any further feedback or final recommendations at your earliest convenience?
> > >
> > > Thank you for your contribution to TMLR
> > >
> > > Best, AE

---

### Review · Reviewer_2eUy · 2026-07-03

**Summary Of Contributions:**

Summary:

The paper proposes WISE (Which-Why Informed Semantic Explorer), a low-level controller framework for long-horizon embodied tasks in Minecraft that augments episodic memory with causal reasoning.

WISE consists of three main components: a Causal Event Graph for causally guided memory retrieval, an Opportunistic Task Scheduler for dynamic task prioritization, and a three-tier multi-scale exploration strategy (quadtree global, frontier-based regional, Voronoi local completion). The retrieval mechanism combines visual similarity with causal matching to improve robustness over feature-based retrieval.

Empirically, WISE reports gains over MrSteve on exploration coverage, sequential sparse tasks (ABA), and adaptive non-sequential tasks (ABC), plus leave-one-out ablations and a VLM cost analysis. The paper claims that causal reasoning improves retrieval robustness, enables adaptive task execution, and that the proposed components exhibit super-additive synergy.

Strengths:

1) Using affordance-style causal reasoning to guide opportunistic online task re-scheduling is a well-motivated idea that distinguishes WISE from fixed-sequence controllers such as MrSteve.

2) The use of semantic, entity-level retrieval instead of relying solely on visual feature similarity provides improved robustness to viewpoint changes and is a practically useful contribution.

3) The overall framework is well designed, with a coherent integration of memory, causal reasoning, scheduling, and exploration. The asynchronous VLM design is a great engineering choice that meets Minecraft's constraints when trying to leverage a large VLM.

Weaknesses:

1) The paper contains several consistency issues that reduce confidence in the experimental results. These include inconsistent GPT-4o versions, slightly confusing GPU system descriptions, an invalid PyTorch version, and anomalies in the exploration coverage tables that appear inconsistent with the claimed O(log n) scaling.

2) The use of the term causal is overstated. The proposed Causal Event Graph does not model interventions P(y|do(x)), structural causal models, or counterfactual reasoning as defined in the Pearl Causal Hierarchy. Instead, the CAN_OBTAIN edges are deterministic affordance relations extracted from external knowledge, while CO_OCCURS_WITH represents simple L1 statistical co-occurrence. The proposed graph is therefore closer to a semantic or affordance graph(Just L1) than a causal graph (L2/L3), and the terminology should be revised or better justified.

3) The synergy claim is not supported by the experimental design. Leave-one-out ablation measures each component's marginal value given the others, but cannot identify interaction/synergy effects as that requires a factorial study with interaction analysis, which is not provided.

4) The central claim that semantic retrieval is more robust to viewpoint changes is not validated by a controlled experiment.

**Audience:**

Yes

**Audience Explanation:**

The question of how to incorporate causal or semantic structure into the memory and control layer of embodied agents is a clear interest to the embodied AI, LLM-agent, and Minecraft benchmark communities. The idea of using causal or affordance relations to guide opportunistic online task re-scheduling in a low-level controller is both novel and well motivated. Furthermore, the emphasis on integrating "why/which" reasoning with episodic memory provides a useful conceptual contribution, even though the current experimental evidence does not fully support all of the paper's claims.

**Broader Impact Concerns:**

No specific broader-impact concerns as the work is confined to agents operating within the Minecraft simulator using only raw visual observations with no human-subjects data.

**Claims And Evidence:**

No

**Claims Explanation:**

1) The experimental evaluation lacks sufficient statistical rigour. Although each task is evaluated over 50 episodes, the experiments appear to use only a single map layout with different starting seeds. The paper does not report standard deviations, confidence intervals, or statistical significance tests. In addition, the priority weights (0.3, 0.5, 0.2) used in the Opportunistic Task Scheduler are described as being "empirically set", but no parameter sweep or sensitivity analysis is provided. Since these weights directly influence task prioritisation and the behaviour of the proposed causal framework, it is important to understand how sensitive the results are to their choice. As presented, it is difficult to determine whether the reported improvements are robust or are influenced by the selected map and parameter settings. The evaluation should therefore include experiments on multiple map configurations together with appropriate statistical analyses and a sensitivity study of the scheduler weights.

2) The paper does not isolate its main proposed mechanism. The claim that semantic retrieval is more robust to viewpoint changes than MineCLIP feature similarity is inferred only from end-to-end task performance, where exploration and scheduling also influence the results. A controlled retrieval experiment under varying viewpoints is needed to directly validate this claim. An important baseline is missing: retrieval using GPT-4o-generated entity labels without the proposed causal graph. Without this comparison, it is not possible to determine whether the reported gains come from the graph structure itself or simply from using a stronger vision-language model.

3) The map coverage results contain several inconsistencies that are not explained. In Table 1a, on the 384×384 simulated map, the Quadtree-Based variant achieves the same coverage as MrSteve (0.83), while the full WISE model reaches 0.98. However, on the smaller map, the Quadtree-Based variant shows a much larger improvement over MrSteve. This is unexpected, as the proposed quadtree strategy is claimed to scale as O(log n), suggesting that its benefit should become more pronounced on larger maps rather than smaller ones. In addition, the presentation of Table 1b is unclear, with ambiguous step-to-map correspondence and the quantities do not behave monotonically with the step budget.

4) From the perspective of the Pearl Causal Hierarchy (PCH), the evaluation is largely confined to Level 1 (association). The proposed Causal Event Graph is constructed from GPT-4o and a Wikipedia-derived knowledge base containing predefined affordance relations and the evaluation tasks are based on these same relations. Consequently, the agent is evaluated on knowledge that is already encoded in its graph, making the demonstrated advantage difficult to separate from prior semantic knowledge. As a result, the experiments do not demonstrate causal reasoning beyond associational knowledge, despite the paper's framing in terms of causality.

5) Consistency issues:

i) PyTorch "v2.5.5" (no such release exists)

ii) GPT-4o version listed as gpt-4o-2024-05-13 and gpt-4o-2024-11-20 in the paper in 2 different places

iii) Does the system have 1 or 4 RTX4080 16GB? "All experiments are conducted on a single server equipped with four NVIDIA RTX 4080 GPUs" vs. "All experiments were conducted on NVIDIA RTX4080 16GB."

iv) Typos Within Figure 6 -> "recognision", "decison";  Typos Within Figure 2 -> "Shrot-Term"; Typos within the paper -> "Problew setting", "Oppotunistic";

v) Figure 1 caption/panel mismatch ("obtain beef" in caption vs. labeled "obtain log"/"obtain water" in figure) and some incomplete bibtex references without years/venues.

**Requested Changes:**

1) Correct or substantiate the synergy claim. Provide a factorial ablation study with explicit interaction-term analysis, or revise the claim.

2) Strengthen the experimental evaluation by reporting statistical measures such as standard deviations, confidence intervals, and significance tests, together with experiments on multiple map layouts and a sensitivity analysis of the scheduler parameters.
Include the missing baseline that performs retrieval using GPT-4o-generated entity labels without the proposed Causal Event Graph, to isolate the contribution of the graph structure.

3) Add a controlled retrieval experiment that varies viewpoint and occlusion while keeping the scene fixed, and directly compare semantic retrieval with MineCLIP feature-based retrieval. This would provide direct evidence for the claimed robustness instead of relying only on end-to-end task performance.

4) Clarify the use of the term causal. If the authors intend to make claims about causal reasoning, the framework should be discussed in the context of the Pearl Causal Hierarchy and supported with evidence beyond associational (Level 1) knowledge. Otherwise, it would be more appropriate to describe the proposed graph as a semantic or affordance-based representation rather than a causal one.

5) Resolve the ambiguities in the experimental tables, particularly the inconsistencies in the map coverage results and their expected monotonic behaviour. In addition, correct the identified typos, inconsistencies, and incomplete references throughout the manuscript.

---

> ### Author Response · Authors · 2026-07-16
> **Rebuttal to Reviewer  2eUy Part 1**
>
> We thank the reviewer for the careful and constructive feedback. We appreciate the positive comments on the motivation of WISE and the integration of memory, semantic structure, opportunistic scheduling, and exploration. We have revised the manuscript to make the claims more precise, strengthen the experimental
> analysis, and fix the consistency issues identified by the reviewer.
>
> **Concern 1: Correct or substantiate the synergy claim. Provide a factorial ablation study with explicit interaction-term analysis, or revise the claim.**
>
> **Response:**
> We thank the reviewer for pointing out this important issue. We agree that our previous use of causal terminology was overly strong and could lead to confusion with formal causal inference. Specifically, WISE does not estimate interventional distributions such as $P(y\mid do(x))$, nor does it involve structural causal models or counterfactual reasoning defined under Pearl’s Causal Hierarchy. The relations encoded in the graph are not discovered through interventions; rather, they are constructed from semantic affordance associations grounded in Minecraft game mechanisms and VLM-assisted scene understanding.
>
> To address this concern, we have revised the terminology throughout the manuscript. The graph is renamed as the **Semantic Affordance Event Graph (SAEG)**. We replace terms including “causal relation,” “causal edge,” “causal retrieval,” and “causal reasoning” with “semantic affordance relation,” “affordance edge,” “affordance-grounded retrieval,” and “affordance-grounded task reasoning,” respectively. We further clarify in the revised manuscript that WISE leverages structured semantic relevance and affordance associations to support task reasoning, rather than performing formal causal discovery, interventional inference, or counterfactual reasoning.
> These revisions better align the terminology with the actual methodology and avoid overstating the theoretical claims of WISE.

---

> > ### Author Response · Authors · 2026-07-16
> > **Rebuttal to Reviewer 2eUy Part 2**
> >
> > **Concern 2: Strengthen the experimental evaluation by reporting statistical measures such as standard deviations, confidence intervals, and significance tests, together with experiments on multiple map layouts and a sensitivity analysis of the scheduler parameters.**
> >
> > **Response:**
> >
> >  We agree that statistical analysis and parameter sensitivity evaluation are important for improving the reliability and reproducibility of our results. In the revised manuscript,  we report raw success counts alongside success rates and present completion steps as mean ± standard deviation over successful episodes, demonstrating that the improvements of WISE are consistent.
> >
> > Regarding map diversity, we clarify that although the sparse task-completion experiments use a fixed $128\times128$ real Minecraft map, the map itself is highly heterogeneous, containing diverse biomes and terrain structures, including mountains, ponds, deserts, and forests, with sparse task-relevant resources distributed across regions. Therefore, it is not a homogeneous layout but a multi-biome environment requiring exploration, memory, and adaptive scheduling. We further evaluate WISE on two map scales ($128\times128$ and $384\times384$) under both simulated and stochastic Minecraft settings to verify scalability.
> >
> > We also add sensitivity analyses for key parameters affecting memory construction, exploration, and scheduling:
> >
> > (1) VLM keyframe processing group size. We vary the default batch size of 16 frames.
> >
> > | VLM batch size  per request           | 12   | 16    | 20    |
> > | ------------------------------------- | ---- | ----- | ----- |
> > | Average response time per Keyframe(s) | 39s  | 44.1s | 51.1s |
> > | Average VLM requests per episode      | 19   | 14    | 11    |
> >
> >
> >
> > The results show a trade-off between update frequency and VLM latency, with the default setting achieving a balance between graph update efficiency and computational cost.
> >
> > (2) Keyframe extraction thresholds.
> >
> > | ($ \theta_c, \theta_H$) | (0.65,0.35) | (0.75,0.25) | (0.85,0.15) | (0.95,0.05) |
> > | ----------------------- | ----------- | ----------- | ----------- | ----------- |
> > | Total Keyframe          | 357         | 281         | 217         | 425         |
> >
> >
> >
> > The default setting selects a compact yet informative keyframe set, while overly permissive selection increases VLM processing cost.
> >
> > (3) Quadtree depth for exploration. We evaluate different spatial decomposition granularities and observe that the default configuration provides the best balance between exploration coverage and efficiency.
> >
> > | Quadtree depth weight | 0.48 | 0.6  | 0.72 |
> > | --------------------- | ---- | ---- | ---- |
> > | 5000timesteps         | 0.56 | 0.71 | 0.52 |
> > | 10000timesteps        | 0.86 | 0.97 | 0.84 |
> >
> > The results indicate that the exploration performance is sensitive to the spatial granularity of the quadtree decomposition. The default setting provides an effective balance between exploration coverage and search efficiency.
> >
> > (4) Scheduler priority weights $(w_u,w_a,w_n)$. We further analyze different weighting configurations, where $w_a$ represents semantic affordance relevance.
> >
> > | $(w_u,w_a,w_n)$ | （0.3，0.5，0.2） | （0.5，0.3，0.2） | （0.3，0.2，0.5） | （0.3，0.4，0.3） |
> > | --------------- | ----------------- | ----------------- | ----------------- | ----------------- |
> > | Success Rate    | 77%               | 68%               | 56%               | 74%               |
> > | Total steps     | 4620              | 5830              | 6650              | 5225              |
> >
> > The results confirm that affordance-aware scheduling improves task efficiency, while extreme weighting choices may lead to less balanced exploration-exploitation behaviors.

---

> > > ### Author Response · Authors · 2026-07-16
> > > **Rebuttal to Reviewer 2eUy Part 3**
> > >
> > > **Concern 3: Include the missing baseline that performs retrieval using GPT-4o-generated entity labels without the proposed Causal Event Graph, to isolate the contribution of the graph structure.**
> > >
> > > **Response:**
> > > We thank the reviewer for this important suggestion. We agree that the previous evaluation did not fully disentangle the contribution of SAEG from that of VLM-based semantic recognition. To address this issue, we add a new baseline, **VLM Entity Retrieval without SAEG**, which uses GPT-4o-generated entity labels for retrieval but does not construct semantic affordance edges or perform graph-based traversal. This baseline isolates the contribution of the proposed SAEG structure. The experimental results are as follows:
> > >
> > > | Method                  | Success Rate | Avg. TimeSteps |
> > > | ----------------------- | ------------ | -------------- |
> > > | WISE w/o graph + prompt | 56%          | 6320           |
> > > | WISE(full model)        | 77%          | 4620           |
> > >
> > > The results confirm that VLM labels help semantic recognition, but SAEG contributes structured causal linking, reusable affordance memory, and more stable task-level retrieval.
> > >
> > > **Concern 4: Add a controlled retrieval experiment that varies viewpoint and occlusion while keeping the scene fixed, and directly compare semantic retrieval with MineCLIP feature-based retrieval. This would provide direct evidence for the claimed robustness instead of relying only on end-to-end task performance.**
> > >
> > > We appreciate this suggestion. We add a controlled retrieval experiment with a fixed Minecraft scene containing a pond target, where viewpoint, distance, and occlusion are varied while keeping the scene unchanged. We compare MineCLIP feature retrieval and GPT-4o semantic retrieval under different target visibility ratios. The results indicate that semantic retrieval is more robust to viewpoint changes and partial occlusions than feature-based retrieval, providing more reliable retrieval signals for downstream task execution.
> > >
> > > | Method                            | 3.6% | 10.2% | 12.4% | 17.3% | 20.8% | 49.4% |
> > > | --------------------------------- | ---- | ----- | ----- | ----- | ----- | ----- |
> > > | MineCLIP feature retrieval        | 0    | 0     | 0     | 0     | 1     | 1     |
> > > | GPT-4o semantic retrieval success | 0    | 1     | 1     | 1     | 1     | 1     |
> > >
> > > Here, the column value denotes the percentage of the image occupied by the pond under different viewpoints or occlusion settings. A value of 1 indicates successful retrieval of the pond target, while 0 indicates retrieval failure.

---

> > > > ### Author Response · Authors · 2026-07-16
> > > > **Rebuttal to Reviewer 2eUy Part 4**
> > > >
> > > > **Concern 5: The map coverage results contain several inconsistencies that are not explained. In Table 1a, on the $384\times384$ simulated map, the Quadtree-Based variant achieves the same coverage as MrSteve (0.83), while the full WISE model reaches 0.98. However, on the smaller map, the Quadtree-Based variant shows a much larger improvement over MrSteve. This is unexpected, as the proposed quadtree strategy is claimed to scale as $O(\log n)$, suggesting that its benefit should become more pronounced on larger maps rather than smaller ones..**
> > > >
> > > > **Response:**
> > > >
> > > > We thank the reviewer for this important observation. We clarify that the $O(\log n)$ claim refers to the computational efficiency of quadtree-based region lookup and next-target selection, rather than directly predicting final coverage improvement. The realized coverage depends not only on target selection, but also on low-level navigation success, terrain structure, travel distance, local exploration, and stochasticity under a fixed step budget.
> > > >
> > > > The quadtree strategy improves global exploration by reducing redundant revisits and selecting unexplored regions more efficiently. However, on the larger $384\times384$ map, more steps are consumed by long-range navigation and local execution, limiting the normalized coverage gain. Therefore, the smaller improvement of the quadtree-only variant on the larger map does not contradict its $O(\log n)$ efficiency advantage. We have clarified this distinction in the revised manuscript.
> > > >
> > > > **Concern 6: In addition, the presentation of Table 1b is unclear, with ambiguous step-to-map correspondence and the quantities do not behave monotonically with the step budget**
> > > >
> > > > **Response:**
> > > >
> > > > We thank the reviewer for pointing out the ambiguity in Table 1b. We clarify that “steps” denote Minecraft environment **timesteps** (20 timesteps correspond to approximately one second in Minecraft) and revise the table header and caption accordingly.
> > > >
> > > > The apparent non-monotonicity is caused by mixing different map sizes: the 5,000- and 10,000-timestep results are for the $128\times128$ map, while the 20,000- and 40,000-timestep results are for the $384\times384$ map. Since coverage is normalized by map area, results across different map sizes should not be interpreted as a monotonic function of timestep. We reorganize the table by map size to make this distinction explicit.
> > > >
> > > > **Concern 7: correct the identified typos, inconsistencies, and incomplete references throughout the manuscript.**
> > > >
> > > > **Response:**
> > > >
> > > > We thank the reviewer for identifying these issues. We have carefully revised the manuscript to improve consistency and correctness. Specifically, we replace the invalid PyTorch version `v2.5.5` with the verified version `v2.5.1`, use the GPT-4o identifier `gpt-4o-2024-05-13` consistently throughout the manuscript, and clarify the hardware setup as: “all experiments were conducted on a single server equipped with `4` NVIDIA RTX 4080 16GB GPUs, with each rollout using `1` GPU.” We also correct all identified typos, revise Figure 1 to align the caption with panel labels, and complete missing BibTeX metadata, including publication years and venues.

---

> > > > > ### Comment · Action_Editor_A2UX · 2026-07-16
> > > > >
> > > > > Dear Reviewer,
> > > > >
> > > > > The authors have now submitted their rebuttal and revised manuscript. Could you please review the revised submission and the authors' responses, and provide any further feedback or final recommendations at your earliest convenience?
> > > > >
> > > > > Thank you for your contribution to TMLR
> > > > >
> > > > > Best, AE